# One Step Towards Sustainable Self-supervised Learning

## Abstract

Although increasingly training-expensive, most self-supervised learning (SSL) models have repeatedly been trained from scratch but not fully utilized, since only a few SOTAs are employed for downstream tasks. In this work, we explore a sustainable SSL framework with two major challenges: i) learning a stronger new SSL model based on the existing pretrained SSL model, also called as " base" model, in a cost-friendly manner, ii) allowing the training of the new model to be compatible with various base models. We propose a Target-Enhanced Conditional (TEC) scheme which introduces two components to the existing mask-reconstruction based SSL. Firstly, we propose patch-relation enhanced targets which enhances the target given by base model and encourages the new model to learn semantic-relation knowledge from the base model by using incomplete inputs. This hardening and target-enhancing help the new model surpass the base model, since they enforce additional patch relation modeling to handle incomplete input. Secondly, we introduce a conditional adapter that adaptively adjusts new model prediction to align with the target of different base models. Extensive experimental results show that our TEC scheme can accelerate the learning speed, and also improve SOTA SSL base models, *e.g.,* MAE and iBOT, taking an explorative step towards sustainable SSL.

## 1 Introduction

Self-supervised learning (SSL) has achieved overwhelming success in unsupervised representation learning, with astonishingly high performance in many downstream tasks like classification (Zhou et al., 2022a;b), object detection, and segmentation (Bao et al., 2021; He et al., 2022). In SSL, a pretext task is first built, *e.g.,* instance discrimination task (He et al., 2020; Chen* et al., 2021) or masked image modeling (MIM) (Bao et al., 2021; He et al., 2022), and then pseudo labels are generated via the pretext task to train a network model without requiring manual labels. Though successful, SSL is developing towards a direction of requiring increasingly large training costs, *e.g.,* 200 training epochs in MoCo (He et al., 2020) while 16,00 epochs in MAE (He et al., 2022) to release its potential. Unfortunately, most researchers only have limited computational budgets and often cannot afford to train large SSL models. Moreover, the pretrained non-SOTA SSL models are rarely used in practice, since SOTA is updated frequently and a previous one quickly becomes useless, wasting huge training resources. Thus, a **sustainable SSL** framework is much demanded.

Just like how human experience is enriched and passed from one generation to the next in human society, we try to let an SSL model inherit the knowledge from a pretrained SSL base model to achieve superior representation learning ability for "sustainable" learning and also to improve learning efficiency than training a new SSL model from scratch. Fig. 1 illustrates the sustainable SSL for more clarity, in which we call the new SSL model to be trained as the new model and the pretrained SSL model as the base model. To surpass the base model, in sustainable SSL, the new

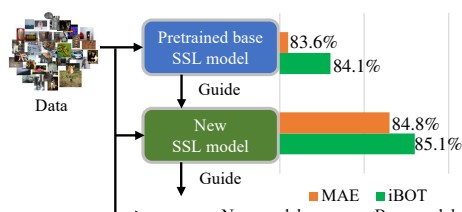

Figure 1: The concept of sustainable SSL.

model exploits not only the implicit base model knowledge but also the absent knowledge in the base model. Such a learning process follows a fully self-supervised manner and differs from the self-training schemes (Xie et al., 2020; Yalniz et al., 2019) that require labels for supervised learn-

ing. This process can be regarded as a special case of Knowledge Distillation (KD) (Hinton et al., 2015; Gou et al., 2021), which targets at learning a more powerful new model based on the base model in a self-supervised setting.

In this work, we take an explorative step towards sustainable SSL by efficiently learning from existing pretrained SSL models and surpassing them. In this work, to achieve this challenging goal, we encourage the new model to learn not only knowledge of the base model but also more semantic-related new knowledge. We therefore choose a mask-reconstruction (He et al., 2022) SSL scheme to train the new model, in which the base model generates reconstruction targets from the full input images and the new model tries to predict the generated targets from randomly masked image input. With this pretext task, the new model is forced to learn the se-

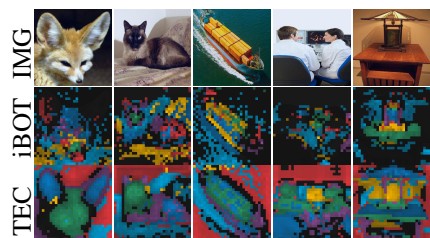

Figure 2: Self-attention visualization. Different colors denote attentions of different heads. Black means no attention.

mantics of the full input and its patch relations so that the new model can reason the desired full information from an incomplete input. As illustrated by Fig. 2, the attentions of iBOT (Zhou et al., 2022a) miss some semantic regions, *e.g.,* ears, while TEC with iBOT as the base model captures all semantics and well distinguishes all different components of an input image. Because of its more powerful ability to capture comprehensive semantic, TEC helps achieve the challenging sustainable SSL, and actually can provide rich and flexible semantics for downstream tasks.

However, different SSL base models could have various properties due to their various training targets and training strategies, *e.g.,* iBOT models with more category semantics while MAE models with more image details (He et al., 2022). So it is important to build high-qualified and compatible reconstruction targets from the base model so that the new model learns these targets in a complementary manner. A good model target should reveal the semantic relations among patches, *e.g.,* the relation between car wheels and car body, so that new model can learn these general relation patterns and adapts to downstream tasks. To this end, we propose to enhance the target quality of the base model by using two complementary reconstruction targets: a) the patch-dim normalization which normalizes base model targets along patch dimension to enhance the relations among input patches, and b) patch attention maps with rich semantics to filter out possible noise and establish the correlation between the whole image semantic and the patch semantic. For target compatibility, we introduce conditional adapters into the new model so that new model predictions can be adaptable to various base models with different properties. Given a base model target, adapters conditionally active and adjust mid-level features of the new model to predict the target more effectively. These adapters are discarded after pretraining but can serve parameter-efficient finetuning (Jia et al., 2022; Chen et al., 2022b) if kept.

We call the above method for sustainable SSL as Target-Enhanced Conditional (TEC) mask-reconstruction. As shown in Fig. 3, on ImageNet, TEC without any extra training data improves the SSL base model by a remarkable margin, *e.g.,* MAE (He et al., 2022) and iBOT (Zhou et al., 2022a). For instance, taking iBOT with 1600 epochs as base model, TEC with only 800 training epochs makes 1.0% improvement. Moreover, we also find that TEC can significantly accelerate the SSL learning process and saves training cost. For example, training TEC for only 100 epochs with random initialization and a 300-epochs-trained MAE base model outperforms MAE trained with 1600 epochs. This work takes one step closer to sustainable SSL, and we hope our initial effort will inspire more works in the future to sustainably improve SSL in a cost-friendly manner.

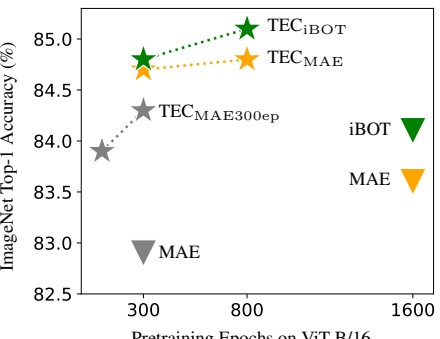

Figure 3: Top1 accuracy on ImageNet-1k. TEC models have the same color with their base model.

Figure 4: The overall framework of the proposed TEC. The pretrained SSL base model in TEC generates patch-relation enhanced reconstruction targets, *i.e.,* patch-dim normalized features and semantic attention maps. The new ViT encoder takes in masked image and the class token enhanced by the input adapter, and then sequentially passes the generated features into encoder adapters and the multi-target decoder to predict the targets given by base model.

## 2 METHOD

### 2.1 OVERALL FRAMEWORK

An overall framework of the proposed target-enhanced conditional (TEC) mask-reconstruction method is illustrated in Fig. 4. TEC follows (He et al., 2022; Bao et al., 2021) to use Vision Transformer (ViT) (Dosovitskiy et al., 2020) for implementation. Under the mask-reconstruction framework (He et al., 2022), as shown in Fig. 4, TEC consists of a new ViT encoder to be pretrained, conditional adapters for conditional pretraining, a multi-target decoder for reconstruction targets prediction, an SSL pretrained ViT encoder as the base model, and a target-enhancing module to construct patch-relation enhanced reconstruction targets from the base model. Specifically, the base model is an SSL-pretrained ViT encoder (*e.g.,* in MAE (He et al., 2022)) and is used to generate the latent semantic of a full image. Then target-enhancing module enhances the latent semantic to construct two complementary reconstruction targets as the supervision of the new model. The new ViT encoder together with adapters takes in masked images, and generates adapted latent semantics that are then fed into the multi-target decoder to predict the base model targets. After pretraining, the new ViT encoder is kept for downstream tasks while other parts are removed. At below, we will introduce the conditional pretraining aided by adapters in Section 2.2 which helps the new model effectively predict base model targets, and elaborate on the target-enhancing module to generate high-qualified base model targets in Section 2.3.

### 2.2 CONDITIONAL PRETRAINING

As aforementioned, base models often have different properties, *e.g.,* more global category semantic in iBOT while more local details in MAE. So the prediction of the new model should be compatible with any given base model. To resolve a similar issue on vanilla image pixel reconstruction, the works (Wang et al., 2022a; Dong et al., 2022; Gao et al., 2022) manually select certain features from the mid-level layers of the encoder by trial and error to better align with the image pixel target. However, it is almost impossible to manually select features from certain fixed layers that are compatible with different base models because of their possible different properties. So to better predict base model targets, the new model should have conditional adaptation ability regarding a given SSL base model.

Given a fixed pretrained model, the parameter-efficient fine-tuning scheme introduces trainable extra modules with a small number of parameters into this pretrained model for adapting it to downstream tasks in both vision (Jia et al., 2022; Chen et al., 2022b) and NLP (Houlsby et al., 2019; Li & Liang, 2021; Liu et al., 2021) domains. For example, the prompting scheme (Li & Liang, 2021; Liu et al., 2021; Jia et al., 2022) concatenates learnable input tokens, *e.g.,* class token, with patch tokens to activate certain semantic features of a fixed ViT model that are suitable for specific downstream tasks. Also, inserting lightweight adapter modules (*e.g.,* MLP (Houlsby et al., 2019; Chen et al., 2022b) and residual blocks (Li et al., 2022b)) into a fixed model can modulate mid-level features of the model to predict features required by the downstream task. Inspired by these parameter-efficient fine-tuning schemes, we introduce the adaptation scheme into the pretraining stage to handle the diversities of base models by equipping the new model with conditional adapters. Since our adapters are only used for pretraining and will be removed during finetuning, they do not increase extra

inference costs. Actually, Tab. 4 shows that keeping these adapters in the inference phase enhances the parameter-efficient finetuning ability of the model. At below, we will introduce how to apply adapters, *i.e.,* input and encoder adapters, into the new model encoder.

**Input adapter.** For ViT networks, one often concatenates a class token with the input patch tokens to learn the global semantics of the whole input. Since the prompting scheme shows the adaption ability of the class token, we propose to further enhance the feature adaption ability of the class token by adding an input adapter. As shown in Fig. 5, the input adapter which is implemented by a small two-layer MLP layer enhances the representation ability of the class token so that the class token can better activate features in the new

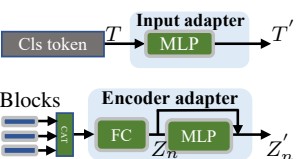

Figure 5: The input adapter and encoder adapter.

model according to the base model targets. Specifically, the class token $T \in \mathbb{R}^C$ of the ViT is processed by the MLP layer to obtain an enhanced class token $T^{'} \in \mathbb{R}^C$:

$$T^{'} = \mathbf{MLP}(T),$$

where $C$ is the embedding dimension. During pretraining, $T^{'}$ is appended to the patch tokens. **MLP** enhances the representation ability of $T$ and enables the new model to better predict base model targets. For inference, since $\mathbf{MLP}(T)$ is shared by all input samples, one can compute it in advance to get $T^{'}$ as the new class token, meaning no extra cost is brought by $\mathbf{MLP}(T)$.

**Encoder adapter.** To modulate mid-level features in the new model so that it can adapt to the base model targets, we apply a simple MLP with residual connection (Chen et al., 2022b) as our encoder adapter in the pretraining phase. As we hope to remove adapters after pretraining for higher inference efficiency, we need to keep the encoder network topology unchanged after removing adapters. So we put the input of adapters in the middle of the encoder and merge all adapter outputs at the end of the encoder. As shown in Fig. 5, given features $X = \{X_i, i = 1, \ldots, D\}$ from each encoder block where $D$ is the encoder block number, we first uniformly divide them into $N$ groups, in which each group contains 3 blocks by default. Within the $n_{th}$ group, we merge features from all blocks:

$$Z_n = \mathbf{FC}(\mathbf{Concat}(X_i, ..., X_j)).$$

Then we feed the feature $Z_n$ into an adapter and obtain an overall feature $Z_e$:

$$Z^{'}_n = Z_n + \mathbf{MLP}(Z_n), \qquad Z_e = \sum_{n=1}^{N} Z^{'}_n, \tag{1}$$

where **MLP** is a small MLP of two fully-connected layers. The adapted features are then fed into the multi-target decoder to predict base model targets, which will be introduced in Section 2.3.

## 2.3 Patch-relation enhanced Reconstruction Targets

To better exploit the knowledge of base models for sustainable SSL, our target-enhancing module constructs two complementary targets with enhanced patch relations: 1) feature-level targets with patch dimension normalization to strengthen the relations among patches; 2) semantic attention maps to learn relations between semantic patches and other patches. The feature-level targets reveal the semantics of certain patches, while attention maps focus more on relations among patch feature.

**Patch-dim normalized feature-level targets.** Given a base model, we propose to normalize its target along the patch dimension to enhance the spatial patch relations. Specifically, for an input, assume its base model target is $Y \in \mathbb{R}^{L \times C}$ where $L$ and $C$ respectively denote patch number and channel dimension. Then we normalize $Y$ along the patch dimension:

$$Y_f = (Y - \mu_L)/\sigma_L, \tag{2}$$

Figure 6: The patch similarity distribution of MAE.

where $\mu_L$ and $\sigma_L$ are respectively mean and variance along the patch dimension. For MIM, this patch-dim normalization can better enhance the spatial relations among tokens than the widely used feature normalization (Wei et al., 2022c;a; Baevski et al., 2022) along channel dimension. This is because as observed in Fig. 6, when using MAE for pretraining, the base model features of different patches actually have similar values and thus have high similarity score, since they all reveal the global semantic of the image. This cannot well reveal the spatial relations among these patches. As a result, the new model can easily reconstruct the feature

target of masked patches due to their high similarity with visible patches, but does not well learn spatial relations among patches. Normalization along channel dimension can hardly enhance the patch relations as it only consider the mean and variance within a patch. Actually, channel-dim normalization even enlarges the similarity among patches as revealed in Fig. 6. In contrast, patch-dim normalization ensures values within each channel has a clear difference and enhances the possible inherent spatial relations among patches by obviously reducing the similarity among patches as shown in Fig. 6. Moreover, Tab. 6(c) shows that our normalization can significantly improves the new model performance. After normalization, following (He et al., 2022), the new model uses a fully-connected layer followed by the decoder to generate $Z_f$ for predicting the base model target $Y_f$ on masked patches:

$$L_{\text{fea}} = \|M \circ (Y_f - Z_f)\|_2^2, \tag{3}$$

where $M$ is the mask matrix and $\circ$ denotes the element-wise product.

**Semantic attention-level targets.** Self-attention in pretrained ViT models has a powerful capability of capturing semantic relations among patch tokens (Caron et al., 2021; Li et al., 2022c; Ziegler & Asano, 2022). We then propose to utilize the self-attention map as a type of reconstruction target for MIM to further enhance the semantic relation modeling capability of the new model. According to previous investigations on the effects of self-attention map in KD (Wu et al., 2022; Wang et al., 2022b), not all attention map contains useful semantic relations, and severe noisy attentions even hinder student learning. Accordingly, it is necessary to select parts of attention maps to reduce the possible severe noise and also help reduce training costs.

Here we utilize the base model class token that contains sufficient global semantics to select top similar patch tokens, which filters out the possible noises. Specifically, given the attention maps $A_c \in \mathbb{R}^{H \times L}$ between class token and patch tokens from the last ViT block in base model where $L$ and $H$ respectively denote patch number and head number, we average the attention map $A_c$ along head dimension to obtain $A_c' \in \mathbb{R}^{1 \times L}$. Then, as shown in Fig. 4, we select top-$k$ patches with the largest values in $A_c'$, and then compute the attention map $A_p \in \mathbb{R}^{H \times k \times L}$ among the top-$k$ patches and all patch tokens. Considering the importance of the class token, we further concatenate attention maps between itself and selected $A_p$ to obtain our final reconstruction targets, *i.e.,* $A_s \in \mathbb{R}^{H \times (k+1) \times L}$. Note, when we compute $A_s$, a temperature $\tau$ is added before the Softmax operation to adjust the attention sharpness. For the new model, we respectively use two fully-connected layers to project its decoder output into two predictions $Z_q \in \mathbb{R}^{L \times C}$ and $Z_k \in \mathbb{R}^{L \times C}$. We select the same patches as in $A_s$ from $Z_q$ to form $Z_q' \in \mathbb{R}^{k \times C}$. Then we concatenate the class token cls in new model with $Z_q'$ and compute the KQ attention map $Z_a = \text{Softmax}([Z_q', \text{cls}]^\top Z_k) \in \mathbb{R}^{H \times (k+1) \times L}$. Finally, we compute the predicted entropy loss between the prediction $Z_a$ and the target $A_s$ as

$$L_{\text{att}} = -A_s \log Z_a. \tag{4}$$

**Multi-target decoder.** Due to the different properties of two reconstruction targets, namely feature target and attention target, one decoder in a new model for prediction is insufficient to handle them simultaneously and often results in prediction conflict. But using separate decoders for each target would increase the trainable parameters and thus slow down the training. To solve this issue, we use a simple decoder adaptation scheme that constructs target-specific inputs and then feeds them into a shared decoder. Specifically, we feed the output feature $Z_e$ (see Eqn. 1) of the new model encoder into a fully-connected layer and then fill the masked tokens with a learnable mask token to obtain $Z_f'$. Then similarly, given $Z_e$, we also use a fully-connected layer and a learnable mask token to obtain $Z_m'$. Next, we respectively feed $Z_f'$ and $Z_m'$ into a shared transformer-based decoder for predicting the feature and attention map targets of the base model. Unlike the large semantic gap between encoder output and vanilla image in MAE, the base model target has similar semantics to the new model predictions. So a shallow 2-layer decoder is enough and better than the 8-layer decoder used in MAE. This design also greatly reduces the training cost.

## 3 EXPERIMENTS

We evaluate our TEC on ImageNet-1k (Deng et al., 2009) by pretraining randomly initialized ViT (Dosovitskiy et al., 2020) with a 16×16 patch size and 224×224 image resolution for 300/800 epochs via AdamW (Loshchilov & Hutter, 2017) of 4,096 batchsize. To ensure the improvement is from TEC, we do not use any explicit/implicit extra training data and the base models that are stronger than new model. Indeed, we respectively use iBOT (Zhou et al., 2022a) and MAE (He

Table 1: Comparison with existing SSL methods under ImageNet-1k finetuning using ViT. † and gray color mean the usage of implicit /explicit extra data. The pretraining epoch number of TEC denotes the one from randomly initialized weights under the guidance of base models, and does not include that of the base model. Compared results are obtained from their reported results.

| Model | Method | Epoch | Guidance | Top1 acc. |
|---|---|---|---|---|
| ViT-Base | Deit III (Touvron et al., 2022) | 800 | Supervised | 83.8 |
| | DINO (Caron et al., 2021) | 300 | NA | 82.8 |
| | MoCov3 (Chen* et al., 2021) | 300 | NA | 83.2 |
| | MixMIM (Liu et al., 2022b) | 300 | RGB | 83.2 |
| | MFM (Xie et al., 2022a) | 300 | Frequency | 83.1 |
| | BEiT (Bao et al., 2021) | 800 | DALLE† | 83.2 |
| | SplitMask (El-Nouby et al., 2021) | 300 | NA | 83.6 |
| | ConMIM (Yi et al., 2022) | 800 | Momentum | 83.7 |
| | SimMIM (Xie et al., 2022b) | 800 | RGB | 83.8 |
| | SIM (Tao et al., 2022) | 1600 | Momentum | 83.8 |
| | CAE (Chen et al., 2022c) | 1600 | DALLE† | 83.9 |
| | MaskFeat (Wei et al., 2022a) | 1600 | HOG | 84.0 |
| | LoMaR (Chen et al., 2022a) | 1600 | RGB | 84.1 |
| | BootMAE (Dong et al., 2022) | 800 | RGB+Momentum | 84.2 |
| | data2vec (Baevski et al., 2022) | 800 | Momentum | 84.2 |
| | Mugs (Zhou et al., 2022b) | 1600 | NA | 84.3 |
| | MVP (Wei et al., 2022b) | 300 | CLIP† | 84.4 |
| | PeCo (Dong et al., 2021) | 800 | Perceptual codebook | 84.5 |
| | CMAE (Huang et al., 2022) | 1600 | RGB | 84.7 |
| | Ge2-AE (Liu et al., 2022a) | 800 | RGB+Frequency | 84.8 |
| | FD-CLIP (Wei et al., 2022c) | 300 | CLIP† | 84.9 |
| | MAE (He et al., 2022) | 1600 | RGB | 83.6 |
| | FD-MAE (Wei et al., 2022c) | 300 | MAE | $83.8_{+0.2}$ |
| | **TEC** | 300 | MAE | $\mathbf{84.7}_{+1.1}$ |
| | **TEC** | 800 | MAE | $\mathbf{84.8}_{+1.2}$ |
| | iBOT-ImageNet-22K | - | Momentum | 84.4 |
| | iBOT (Zhou et al., 2022a) | 1600 | Momentum | 84.1 |
| | SemMAE (Li et al., 2022a) | 800 | iBOT | $84.5_{+0.4}$ |
| | **TEC** | 300 | iBOT | $\mathbf{84.8}_{+0.7}$ |
| | **TEC** | 800 | iBOT | $\mathbf{85.1}_{+1.0}$ |
| ViT-Large | MAE (He et al., 2022) | 1600 | RGB | 85.9 |
| | **TEC** | 300 | MAE | $\mathbf{86.5}_{+0.6}$ |

Table 2: Semantic segmentation on ADE20k using Upernet and ViT-B.

| Method | Epoch | mIoU |
|---|---|---|
| BEiT | 800 | 47.1 |
| PeCo | 800 | 48.5 |
| GE2-AE | 800 | 48.9 |
| CAE | 1600 | 50.2 |
| CMAE | 1600 | 50.1 |
| MAE | 1600 | 48.1 |
| **TEC**$_{MAE}$ | 800 | **49.9** |
| iBOT | 1600 | 50.0 |
| **TEC**$_{iBOT}$ | 800 | **51.0** |

Table 3: Instance segmentation on COCO using Cascade MaskRCNN and ViT-B.

| Method | $AP_{bbox}$ | $AP_{mask}$ |
|---|---|---|
| Implementation from (Zhou et al., 2022a) | | |
| iBOT | 51.2 | 44.2 |
| **TEC**$_{iBOT}$ | **52.7** | **45.4** |
| Implementation from (Li et al., 2022b) | | |
| MAE | 54.0 | 46.7 |
| **TEC**$_{MAE}$ | **54.6** | **47.2** |

et al., 2022) pretrained ViT model on ImageNet-1k as our base model. Base models are obtained from their official publicly released versions. We use the same masking strategy in MAE, *e.g.,* 75% random masked ratio. See more training details in Appendix.

## 3.1 PERFORMANCE COMPARISON

### 3.1.1 COMPARISON ON IMAGENET

**Finetuning on ImageNet-1k.** Tab. 1 summarizes the finetuning performance on ImageNet-1k. One can observe that with iBOT as base model, TEC surpasses the base model by 0.7% under 300 training epochs from random initialization, and further makes 1.0% improvement for 800 epochs. Similarly, TEC respectively brings 1.1% and 1.2% improvement over the MAE base model under 300/800 training epochs. These results show that thanks to the proposed target-enhanced conditional MIM scheme, TEC actually can further improve the strong MIM-based methods, *e.g.,* MAE and iBOT used here. Besides, Tab. 1 also shows that under similar or even less training cost, TEC outperforms other SOTA SSLs, including methods trained by implicitly extra data, *e.g.,* MVP (Wei et al., 2022b)

Table 4: Top1 accuracy on the ImageNet-1k dataset under parameter-efficient finetuning.

| Method | Epoch | Settings | Top 1 acc. |
|--------|-------|----------|------------|
| MAE | 1600 | Linear probing | 68.0 |
| TEC$_{\text{MAE}}$ | 800 | Linear probing | 69.8 |
| | | +Input adapter FT | 72.6 |
| | | +Encoder adapter FT | 79.9 |

Table 5: Semi-supervised semantic segmentation on the ImageNet-S dataset.

| Pretrain | Method | Epoch | mIoU$_{\text{val}}$ |
|----------|--------|-------|---------------------|
| SSL | MAE | 1600 | 38.3 |
| | TEC$_{\text{MAE}}$ | 800 | **42.9** |
| SSL+FT | MAE | 1600+100 | 61.0 |
| | TEC$_{\text{MAE}}$ | 800+100 | **62.0** |

and FD-CLIP (Wei et al., 2022c). More surprisingly, TEC with only ImageNet-1k data has an improvement of 0.7% over iBOT trained on ImageNet-22k, indicating more effectiveness of TEC pretraining over more training data. To the best of our knowledge, *TEC sets a new SOTA 85.1% on ViT-B when solely using ImageNet-1k*, showing the potential of sustainable SSL learning. We also investigate the scaling ability of TEC by using ViT-Large, and observe that TEC surpasses the MAE pretrained base model by 0.6% with 300 epochs from random initialization.

**Parameter-efficient finetuning on ImageNet-1k.** Parameter-efficient finetuning methods, *e.g.,* linear probing, aims to finetune a small amount of parameters for adapting to a downstream task. We test TEC under the linear probing setting which only finetunes a linear classifier at the top of a frozen pretrained model. Tab. 4 reports the classification accuracy of ViT-B on ImageNet-1k under linear probing. TEC improves the MAE base model by 1.8%, showing more category-related semantic information in the learned new model. Indeed, our input and encoder adapters used for pretraining can also be used for parameter-efficient finetuning. Finetuning input adapter via prompting brings a remarkable improvement of 4.6%, and finetuning both input and encoder adapters makes 11.9% improvement over the MAE base model. This also shows the benefit of our proposed adapters.

**Semantic segmentation on ImageNet-S.** To test the pixel-level representation ability of TEC pretrained models, we conduct semantic segmentation finetuning on ImageNet-S (Gao et al., 2021) which has pixel-level training labels on ImageNet. We use ViT-B as the segmentation model without extra segmentation head, since the pretraining and finetuning data have no domain shift. Tab. 5 shows that TEC$_{\text{MAE}}$ improves MAE base model by 4.6% on mIoU. When using supervised ImageNet fully-finetuned pretraining, TEC$_{\text{MAE}}$ achieves a gain of 1.0% over MAE.

### 3.1.2 TRANSFER LEARNING ON DOWNSTREAM TASKS

Here we investigate the transfer learning ability of TEC models on downstream tasks.

**Semantic segmentation.** For semantic segmentation on the ADE20k (Zhou et al., 2018) dataset, we use Upernet (Xiao et al., 2018) with ViT-B as the segmentation model. Tab. 2 shows that TEC$_{\text{iBOT}}$ surpasses the iBOT base model by 1.0% mIoU, and TEC$_{\text{MAE}}$ achieves a 1.8% improvement over its MAE base model. Thus, TEC pretrained models show greater transfer learning abilities on semantic segmentation compared to their base models. Also, TEC shows remarkable advantages over strong competitors with fewer pretraining epochs. For example, it outperforms MAE, CAE (Chen et al., 2022c), and CMAE (Huang et al., 2022) by 2.9%, 0.8%, and 0.9%, achieving new SOTA.

**Instance segmentation.** For instance segmentation on COCO (Lin et al., 2014), for fairness, we apply the Cascade MaskRCNN (Cai & Vasconcelos, 2019) implemented by iBOT (Zhou et al., 2022a) and ViTDet (Li et al., 2022b) for TEC with iBOT/MAE base models. Tab. 3 shows that with the implementation from iBOT, TEC surpasses the iBOT base model by 1.5% on box AP and 1.2% on mask AP; and when using the implementation from ViTDet, TEC also achieves a gain of 0.6% on box AP and 0.5% on mask AP, indicating stable improvements of TEC.

### 3.2 ABLATION AND ANALYSIS

We give the ablation study and analysis of our TEC. By default, models are pretrained with 300 epochs and evaluated with the finetuning on ImageNet-1k.

**Conditional pretraining.** The conditional adapters aid the SSL pretraining under different base models. Tab. 6(a) shows adapters stably improve the performance by 0.4% and 0.2% when using MAE and iBOT as base models. To observe the adaptation difference to base models, we show the average proportion of encoder adapters contributing to the encoder output in Fig. 7, *i.e.,* $Z_n^{'}/Z_e$ in Eqn. 1. iBOT base model requires adapters to provide more features

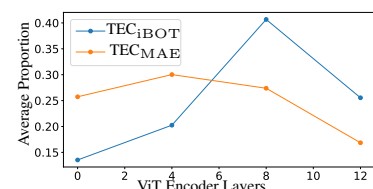

Figure 7: The average proportion of encoder adapters contributing to the encoder output $Z_e$.

Table 6: Ablation study on ImageNet-1K fully finetuning setting using ViT-B.

(a) Ablation study of proposed modules.

| Patch-norm. feature | Semantic attention | Adapters | MAE base | iBOT base |
|---|---|---|---|---|
| Base model performance | | | 83.6% | 84.1% |
| ✓ | | | 84.2% | 84.5% |
| ✓ | ✓ | | 84.3% | 84.7% |
| ✓ | | ✓ | 84.6% | 84.7% |
| ✓ | ✓ | ✓ | 84.7% | 84.8% |

(b) Effect of adapters.

| | Top1 acc. |
|---|---|
| MAE base | 83.6 |
| No adapter | 84.2 |
| + input adapter | 84.3 |
| + encoder adapter | 84.6 |

(c) Patch-norm features.

| | Top1 acc. |
|---|---|
| MAE base | 83.6 |
| NA | 83.9 |
| Feature dim. | 83.9 |
| **Patch dim.** | **84.2** |

(d) Init. with base model pretrain.

| | Top1 acc. |
|---|---|
| iBOT base | 84.1 |
| Load | 84.4 |
| **Not load** | **84.8** |

(e) TEC accelerates MAE training.

| | Epoch | Top1 acc. |
|---|---|---|
| MAE | 1600 | 83.6 |
| $\text{TEC}_{\text{MAE1600ep}}$ | 300 | $\textbf{84.7}_{+1.1}$ |
| MAE | 300 | 82.9 |
| $\text{TEC}_{\text{MAE300ep}}$ | **100** | $\textbf{83.9}_{+1.0}$ |
| $\text{TEC}_{\text{MAE300ep}}$ | 300 | $\textbf{84.3}_{+1.4}$ |

(f) Effect of semantic-related patch attention.

| | Top1 acc. |
|---|---|
| iBOT base | 84.1 |
| No attention | 84.5 |
| Cls token only | 84.5 |
| All attention | 84.6 |
| **Attention select** | **84.7** |

from deeper layers, while MAE base model makes adapters focus more on shallow layers, which is constant with their properties, *i.e.,* iBOT base model has more high-level category semantics while MAE model has more low-level image details.

**Feature normalization on different dimensions.** We normalize target features on the patch dimension to stress the relative relations among patches, which differs from existing methods that normalize features on the channel dimension. In Tab. 6(c), normalizing on patch dimension achieves a 0.3% gain than channel-dim normalization. In contrast, the channel-dim normalization has no effect compared to the unnormalized version. Channel-dim normalization emphasizes the feature difference in channels. Instead, our patch-dim normalization stresses the relations among patches, which is compatible with the patch prediction in the MIM scheme. Tab. 6(a) shows training with patch-dim normalized feature has the 0.6%/0.4% gain over MAE/iBOT base models, showing its robustness over base models.

**Semantic-related attention.** The KQ attention maps naturally contain the semantic relations among patches and thus are used as the base model targets with enhanced patch-relation properties. Tab. 6(a) shows that using attention maps further improves the models trained with patch-dim normalization. Tab. 6(f) compares the effects of different types of attention maps. Only using the attention maps of the class token has no improvement, while the attention of semantic-related patches brings a 0.2% gain over the baseline. Therefore, it is the relation among patches that helps the MIM training. Compared to using all attention maps, using the selected semantic-related attention maps brings a larger gain by reducing the noise.

**Accelerating the training process of base models.** By default, we use the fully pretrained SSL models as base models. To verify if TEC can improve an unconverged SSL model, we use a 300-epoch MAE pretrained ViT-B as the base model and train TEC with 100/300 epoch from random initialization. As shown in Tab. 6(e), the 300-epoch pretrained MAE gives a 82.9% Top.1 accuracy. In comparison, $\text{TEC}_{\text{MAE300ep}}$ achieves 84.3%/83.9% with 300/100 epochs, surpassing the 300-epoch MAE base models with 1.4%/1.0%. Notably, $\text{TEC}_{\text{MAE300ep}}$ even outperforms the 1600-epoch pretrained MAE by 0.3% with only 100 epoch training, showing TEC can significantly accelerate the training process of the base model. Still, the $\text{TEC}_{\text{MAE1600ep}}$ taught with 1600-epoch MAE base model further improves the $\text{TEC}_{\text{MAE300ep}}$ by 0.4%, indicating our sustainable learning scheme relies on good base models to achieve better performance.

**Initializing new model with base model weights or not.** The new models in the TEC framework are trained from random initialization. Tab. 6(d) compares the new model performance with/without loading the pretrained weights of the base model. The randomly initialized new model outperforms the model loaded with pretrained base model weights by 0.4%. We assume that randomly initialized

new models avoid the local minima of the base model, and the new model learns a different weight distribution from the base model.

## 4 RELATED WORKS

**Self-supervised learning.** Self-supervised learning enables representation pretraining without human annotation by training with pretext tasks, *e.g.,* instance discrimination task (ID) and masked image modeling task (MIM). ID learn high category-related representations by pulling close representations from multiple views of one image Chen et al. (2020); Grill et al. (2020); Chen & He (2021); Zbontar et al. (2021); Caron et al. (2020). Extracting representations from multi-view requires larger training cost than supervised training. MIM learns semantics by reconstructing the information of masked patches from unmasked parts, which learns more spatial semantic details than ID. Due to the incomplete input, MIM usually requires longer training epochs than ID to converge. (Huang et al., 2022; Wang et al., 2022a) explore combining the advantages of MIM and ID to further improve performance. Recently, (Kong & Zhang, 2022) reveals both MIM and ID learn occlusion-invariant features. We observe a trend that these SSL methods require increasingly large computing costs to achieve SOTA, which hinders the development of new SSL methods. To remedy this, we explore sustainable SSL by learning from pretrained SSL models.

**Masked image modeling on various targets.** The reconstruction targets guide the MIM learning on different semantic spaces. MIM has explored various reconstruction targets, *e.g.,* RGB pixels and tokenizers. To make images similar to the discretized language in NLP (Devlin et al., 2018), Beit (Bao et al., 2021) utilizes the DALLE pretrained tokenizer (Ramesh et al., 2021) as the prediction target. CAE (Chen et al., 2022c) further decouples this pretext task prediction with the encoder. MAE and SimMIM (Xie et al., 2022b) show using RGB images as targets achieves competitive fully finetuning performance. MaskFeat (Wei et al., 2022a) reveals hand-designed HOG feature (Dalal & Triggs, 2005) is an effective target form. Ge2-AE (Liu et al., 2022a) and MFM (Xie et al., 2022a) find the image frequency domain can be complementary to RGB image targets. The perceptual codebook in PeCo (Dong et al., 2021) helps the model learn semantic information. The online momentum network (He et al., 2020) is used by iBOT and data2vec (Baevski et al., 2022) to provide updated prediction targets. BootMAE (Dong et al., 2022) takes advantage of RGB images and online network targets. (Yang et al., 2022) enhances the distillation from a large teacher model to a compact student model with the masking scheme. MVP (Wei et al., 2022b) introduces rich semantics learned from vision-language pretraining using the CLIP pretrained model (Ramesh et al., 2021) as the target. Unlike these works that stress the unique properties of a specific reconstruction target, we show that all SSL pretrained models can serve as good base models with the help of target-enhancement in TEC. The adapters and target-enhancing scheme in TEC enables the good adaptability to various base model targets.

**Self-supervised knowledge distillation.** The sustainable SSL can be regarded as a special case of the self-supervised knowledge distillation as they both learn from SSL pretrained models. Reversed KD (Yuan et al., 2020) shows a weak teacher model can benefit the student in the supervised setting. ClusterFit (Yan et al., 2020) conducts training on the clustered pseudo-labels to reduce the overfitting to the pretext tasks. SEED (Fang et al., 2021) distillates knowledge from large SSL models to small models with contrastive loss. (Navaneet et al., 2022) uses MLP heads for feature regression to distill large SSL teachers to compact student models. (Xu et al., 2021) groups instances with teacher models and transfers the instance-relation knowledge to student models. As an exception, (Wei et al., 2022c) shows feature distillation improves contrastive-based SSL models but brings marginal gain over the SOTA MAE (He et al., 2022) model. Our sustainable SSL focuses on the new model outperforming the base model in a self-supervised manner. We show in Appendix that our TEC method is advantageous over several SOTA self-supervised distillation methods.

## 5 CONCLUSION

This work explores sustainable self-supervised learning by learning from pretrained SSL models. We propose a target-enhanced conditional mask-reconstruction learning scheme to learn from and surpass existing SSL models. The adapters help to adapt the new model to various base models during pretraining and can also serve as parameter-efficient finetuning modules. We utilize the mask-reconstruction scheme as the basis for surpassing base models, and we construct prediction targets with enhanced patch-level relations to aid the MIM pretraining. Our method further improves the strong MIM pretrained methods, *e.g.,* MAE and iBOT, proving the feasibility of sustainable learning. This work takes an initial step towards sustainable SSL, and we will explore a more general multi-round sustainable SSL framework in the future.

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

# A   APPENDIX

**Towards general sustainable SSL.** This work aims to take one step towards sustainable SSL based on existing pretrained SSL models. To make more steps towards sustainable SSL, we use the TEC pretrained models as the base model for a new round of TEC pretraining. Tab. 7 shows that the second-round TEC trained with the first-round TEC base model achieves 85.2%. The possible reason for the smaller improvement in the second round is caused by the limited network capacity or two rounds of TEC pretraining learns similar knowledge.

**Training cost comparison.**   Accelerating the training of a larger language model with the help of a smaller pretrained model (Qin et al., 2021; Chen et al., 2021) has been proven possible in the NLP field. We follow them to add the FLOPs, training time, and parameters comparison as shown in Tab. 8. TEC requires shorter training time to achieve better performance than the base models. For example, TEC outperforms iBOT/MAE by 0.7%/1.1% Top 1 accuracy with only 7%/20% training time. TEC has a similar number of parameters with MAE, because the shallow decoder saves parameters while adapters increase parameters. Both TEC and MAE have a larger number of parameters compared to iBOT due to the extra decoder. But benefiting from the decoder, they only process the visible patches in the encoder, thus requiring a smaller training cost than iBOT. As only a part of the model requires gradients in some SSL methods, e.g. the base model in TEC and online model in iBOT, which requires no backward cost to compute gradients, we compare the FLOPs for network parts with/without gradients. Benefiting from only processing the unmasked patches in the encoder and the shallow two-layer decoder, TEC requires smaller training FLOPs with gradients than iBOT and MAE. The extra FLOPs (FLOPs without gradients) of the base model in TEC are smaller than the online network in iBOT because no extra head is needed for the base model in TEC. Compared to MAE, the extra FLOPs of the base model can be partly balanced by the smaller FLOPs with gradients in TEC. Therefore, TEC has a similar training time with MAE for each training iteration.

**Comparison on parameter-efficient finetuning.**   Tab. 9 reports the accuracy of 1) learning probing and 2) adapter finetuning which only finetunes the adapter and the linear classifier for parameter-efficient finetuning. One can observe that 1) the linear probing performance of TEC relies on the base model, and 2) adapter finetuning significantly improves the performance. Indeed, most MIM-based models, e.g. BEIT and MAE, have much lower linear probing performance, since they do not use the global semantic learning loss, e.g. clustering loss or InforNCE instance discriminative loss. This also explains the lower performance of TEC compared with the global semantic learning methods, e.g. iBOT. But by finetuning the adaptors and also linear classifier, TEC improves iBOT with a remarkable margin of 3.9%. This is because as shown in Fig. 2, iBOT focuses more on distinguishing the patches related to global semantics and ignores the semantics of other patches, while TEC can group the patches into several semantic groups and further identify the semantics of each group. In this way, finetuning adapters help activate the semantic groups that are related to global semantics required by the downstream tasks, thus improving the model's discriminability on global semantics and showing good parameter-efficient finetuning performance.

**Detailed schematic diagrams.**   We show more detailed schematic diagrams of conditional adapters in Fig. 10 and attention selection in Fig. 11 for a better understanding of these modules.

**Comparison with self-supervised distillation methods.**   We compare several recently proposed self-supervised distillation methods on the fully finetuning performance on ImageNet. Tab. 10 shows the remarkable improvement of TEC over other self-supervised distillation methods. For using MAE ViT-B as the base model, TEC outperforms FD by a noticeable gain of 0.9%. When comparing with MaskFeat which also applies the MIM scheme, TEC has a gain of 0.6% when using MoCov3 ViT-B as the base model.

Table 7: Towards general sustainable SSL using the TEC as the new base model.

| Model | Base | Epoch | Top1 acc. |
|---|---|---|---|
| iBOT | - | 1600 | 84.1 |
| TEC$_{iBOT}$ | iBOT | 800 | 85.1 |
| TEC | TEC$_{iBOT}$ | 800 | 85.2 |

Table 8: Training cost comparison.

| Method | Epoch | Time (8xA100) | FLOPs (with grad) | FLOPs (no grad) | Parameters | Top 1 acc. |
|---|---|---|---|---|---|---|
| VIT-B | - | - | 17.6G | - | 86.6M | - |
| iBOT | 1600 | 361h | 19.2G | 19.2G | 96.3M | 84.1 |
| $\text{TEC}_{\text{iBOT}}$ | 300 | 25h | 8.3G | 17.6G | 118.6M | 84.8 |
| MAE | 1600 | 125h | 9.8G | 0G | 111.9M | 83.6 |
| $\text{TEC}_{\text{MAE}}$ | 300 | 25h | 8.3G | 17.6G | 118.6M | 84.7 |

Table 9: Top1 accuracy on the ImageNet-1k dataset under linear probing (LP), adapter finetuning (Adapter FT), and fully finetuning (Fully FT).

| Method | Epoch | Settings | Top 1 acc. | Fully FT Top 1 acc. |
|---|---|---|---|---|
| BEiT | 800 | LP | 56.7 | 83.2 |
| SimMIM | 800 | LP | 56.7 | 83.8 |
| BootMAE | 800 | LP | 66.1 | 84.2 |
| CAE | 800 | LP | 68.6 | 83.8 |
| SemMAE | 800 | LP | 68.7 | 84.5 |
| CMAE | 800 | LP | 73.9 | 84.7 |
| Ge2-AE | 800 | LP | 75.3 | 84.8 |
| MAE | 1600 | LP | 68.0 | 83.6 |
| $\text{TEC}_{\text{MAE}}$ | 800 | LP | 69.8 | 84.7 |
| $\text{TEC}_{\text{MAE}}$ | 800 | Adapter FT | 79.9 | 84.7 |
| iBOT | 1600 | LP | 79.8 | 84.1 |
| $\text{TEC}_{\text{iBOT}}$ | 800 | LP | 78.0 | 84.8 |
| $\text{TEC}_{\text{iBOT}}$ | 800 | Adapter FT | **81.9** | **85.1** |

**Visualization of patch-dim normalized feature-level targets and semantic attention-level targets.** We visualize the selected semantic attention-level targets from the iBOT base model in Fig. 8. The averaged attention maps of the class token ($A_c^{'}$) mostly focus on the high-semantic objects, thus making the selected patches belong to the high-semantic objects. The attention maps of selected patches contain the semantic relation between high-semantic objects and other regions. Different patches have some unique attention parts that differ from other patches. The attention maps of these selected patches focus on similar semantic objects but are complementary in some parts, which explains why using attention maps of selected patches is better than only using class token attention maps as shown in Tab. 6(f). We show the visualization of patch-dim normalized feature-level targets of iBOT base model in Fig. 9. Patch-dim normalized features are more distinguishable compared to the original and channel-dim normalized features. The spatial relation among feature patches is more clearly shown by the patch-dim normalization.

**Pretraining settings on ImageNet-1k.** We use the standard ViT network implemented in MAE. We give the pretraining settings in Tab. 11, which follows the pretraining settings in MAE. Due to the different properties of SSL base models, we set different parameters of semantic-related attention for MAE and iBOT base models, as shown in Tab. 12.

Inspired by (Baevski et al., 2022), we utilize the average output features of last two blocks of the base model as the feature target. We measure the CKA similarity (Kornblith et al., 2019) of each mid-layer block to the output of the last block, and we observe the high feature similarity of last two blocks.

Table 10: Comparison with self-supervised distillation methods.

| Method | Base | Arch | Epoch | Top 1 acc. |
|---|---|---|---|---|
| MAE | - | ViT-B | 1600 | 83.6 |
| $\text{FD}_{\text{MAE}}$ | MAE-ViT-B | ViT-B | 300 | 83.8 |
| $\text{TEC}_{\text{MAE}}$ | MAE-ViT-B | ViT-B | 300 | 84.7 |
| MoCov3 | - | ViT-B | 300 | 83.2 |
| $\text{MaskFeat}_{\text{MoCov3}}$ | MoCov3-ViT-B | ViT-B | 300 | 83.9 |
| $\text{TEC}_{\text{MoCov3}}$ | MoCov3-ViT-B | ViT-B | 300 | 84.5 |

Image $A_c^{'}$     Multi-head attention maps of class token, top-3 patch, and top-9 patch.

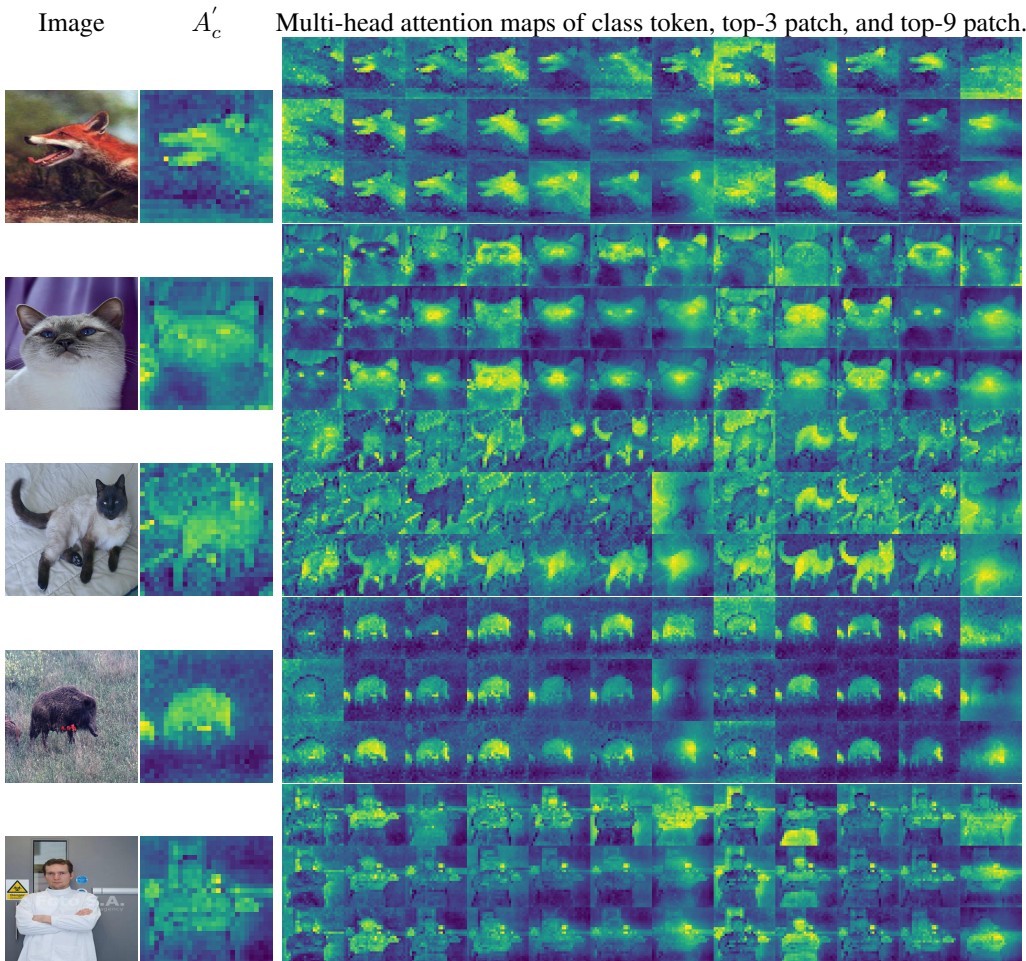

Figure 8: Visualization of the selected semantic attention-level targets from the iBOT base model.

**Fully finetuning settings on ImageNet-1k.** We give the fully finetuning settings on ImageNet-1k in Tab. 13. We observe that TECs trained with different base models may have different properties. Since these base models have different finetuning layer decay values, we set different layer decay values for TECs trained with different base models.

**Parameter-efficient finetuning settings on ImageNet-1k.** Following MAE, the linear probing settings are shown in Tab. 14. For parameter-efficient finetuning with the input adapter, we use the same training settings as used by the liner probing in Tab. 14. When finetuning with the encoder adapters, we use the same training settings as used by the fully finetuning in Tab. 13 due to more parameters are contained in encoder adapters.

**Semi-supervised semantic segmentation finetuning on ImageNet-S.** We give the training settings of semi-supervised semantic segmentation finetuning on ImageNet-S in Tab. 15. We set different learning rates and layer decay weights for models initialized with pretrained weights with/without fully finetuning.

**Downstream task settings.** For semantic segmentation on ADE20K, we use the MMSegmentation (Contributors, 2020) implementation of Upernet. The training configurations follow the MAE training configurations in MMSegmentation. Specifically, the models are trained for 160k iterations with the batch-size of 16 on 8 GPUs. The AdamW optimizer is used with the initial learning rate of 1e-4, weight decay of 0.05, 1,500 warmup iterations, and poly learning rate decay schedule. The ViT-B with 16×16 patch size is used as the backbone. The image size is set to 512×512 and the

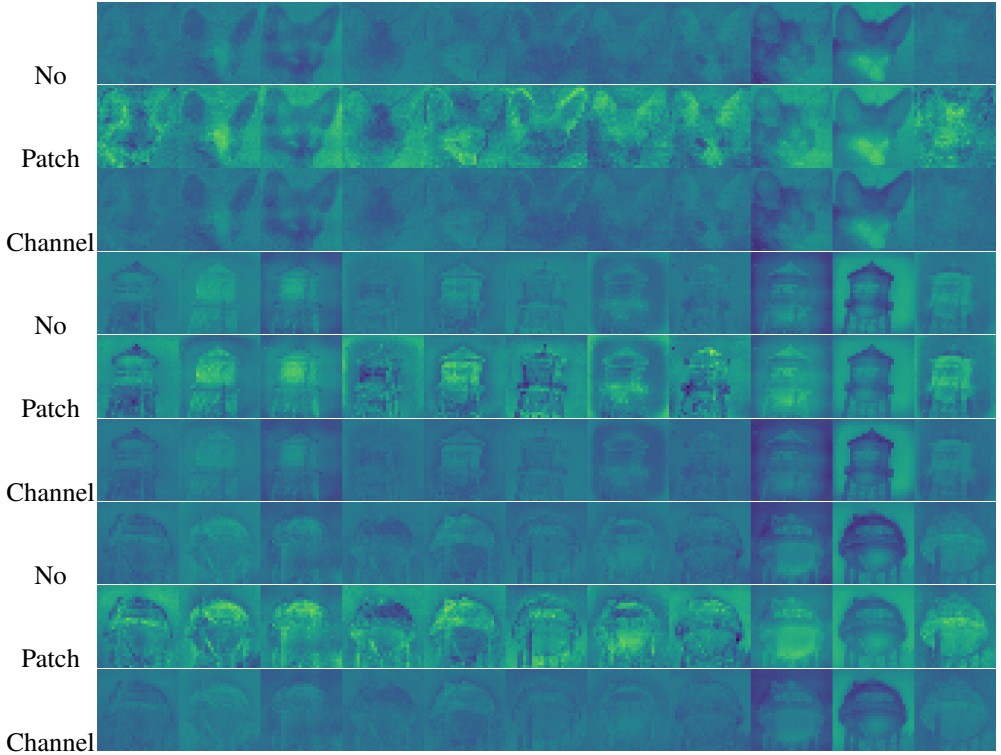

Figure 9: Visualization of patch-dim normalized feature-level targets from the iBOT base model.

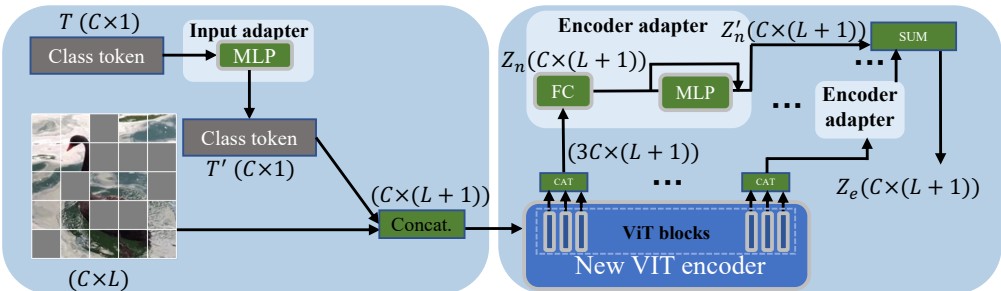

Figure 10: The details of adapters for conditional pretraining.

default data augmentation in MMSegmentation is utilized, *i.e.,* random crop, random flip, and photo metric distortion. A layer decay rate of 0.65 is utilized for the ViT backbone.

For instance segmentation on COCO using Cascade MaskRCNN and ViT-B, we follow the training configurations of iBOT and ViTDet for $TEC_{iBOT}$ and $TEC_{MAE}$. The ViT-B with $16 \times 16$ patch size is used as the backbone. $TEC_{iBOT}$ follows the training strategy of iBOT using the MMDetection implementation. We train the model with the $3\times$ schedule using the batch-size of 16 on 8 GPUs. The AdamW optimizer is used with the initial learning rate of 1e-4, weight decay of 0.05, and layer decay of 0.65. The learning rate is multiplied by 0.1 at the 27th and 33rd epoch. We train the model with $512 \times 512$ image size. The random flip and random resize crop augmentation is applied. For $TEC_{MAE}$, we use the ViTDet implementation of Cascade MaskRCNN. The model is trained with 100 epoch with the batch-size of 64 on 32 GPUs. The AdamW optimizer is used with the initial learning rate of 1e-4, weight decay of 0.1, and layer decay of 0.6. The learning rate is multiplied by 0.1 at 89th and 96th epoch. The input image size is $1024 \times 1024$. The random flip and random resize crop augmentation is applied.

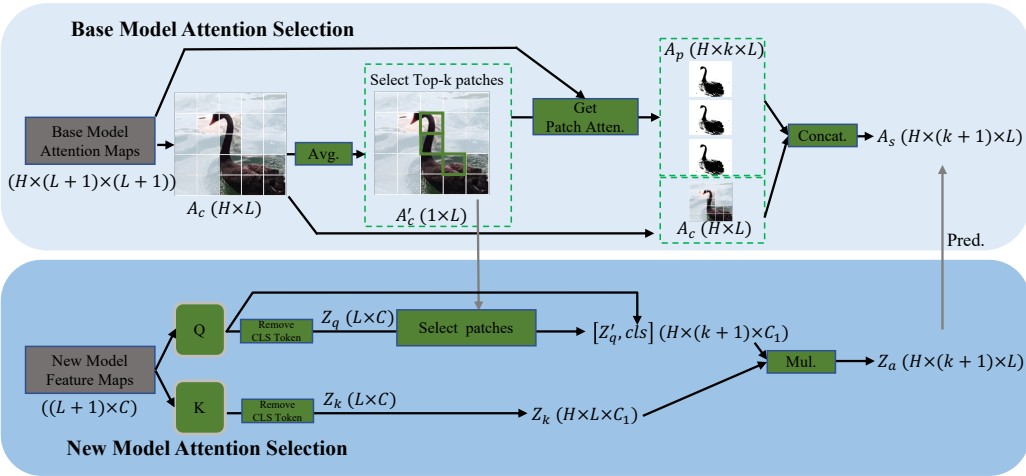

Figure 11: The details of the semantic attention map selection.

Table 11: Pretraining settings.

| Configuration | Value |
|---|---|
| Optimizer | AdamW |
| Base learning rate | 1.5e-4 |
| Weight decay | 0.05 |
| Optimizer momentum | $\beta_1, \beta_2 = 0.9, 0.95$ |
| Batch size | 4096 |
| Learning rate schedule | Cosine decay |
| Warmup epochs | 40 |
| Augmentation | RandomResizedCrop |

Table 12: Parameters of semantic-related attention.

| Base model | $\tau$ | k |
|---|---|---|
| MAE (ViT-Base) | 1.8 | 15 |
| MAE (ViT-Large) | 1.4 | 15 |
| iBOT (ViT-Base) | 1.0 | 9 |

Table 13: Settings of fully finetuning and parameter-efficient finetuning with encoder adapters.

| Configuration | Value |
|---|---|
| Optimizer | AdamW |
| Base learning rate | 5e-4 (B), 1e-3(L) |
| Min learning rate | 1e-6 (B), 1e-5(L) |
| Weight decay | 0.05 |
| Optimizer momentum | $\beta_1, \beta_2 = 0.9, 0.999$ |
| Layer-wise lr decay | 0.55 (MAE-B), 0.65 (iBOT-B), 0.65 (MAE-L) |
| Batch size | 1024 |
| Learning rate schedule | Cosine decay |
| Warmup epochs | 20 (B), 5 (L) |
| Training epochs | 100 (B), 50 (L) |
| Augmentation | RandAug (9, 0.5) |
| Label smoothing | 0.1 |
| Mixup | 0.8 |
| Cutmix | 1.0 |
| Drop path | 0.1 |

Table 14: Settings of linear probing and parameter-efficient finetuning with the input adapter.

| Configuration | Value |
|---|---|
| Optimizer | LARS |
| Base learning rate | 0.1 |
| Weight decay | 0 |
| Optimizer momentum | 0.9 |
| Batch size | 16384 |
| Learning rate schedule | Cosine decay |
| Warmup epochs | 10 |
| Training epochs | 90 |
| Augmentation | RandomResizedCrop |

Table 15: Settings of semantic segmentation finetuning on ImageNet-S.

| Configuration | Value |
|---|---|
| Optimizer | AdamW |
| Base learning rate | 5e-4 (SSL), 1e-4 (SSL+FT) |
| Weight decay | 0.05 |
| Optimizer momentum | $\beta_1, \beta_2 = 0.9, 0.999$ |
| Layer-wise lr decay | 0.60 (SSL), 0.45 (SSL+FT) |
| Batch size | 256 |
| Learning rate schedule | Cosine decay |
| Warmup epochs | 5 |
| Training epochs | 100 |
| Augmentation | RandomResizedCrop |
| Drop path | 0.1 |

