# OpenReview forum: "Towards Sustainable Self-supervised Learning"
_ICLR.cc/2023/Conference — Submitted to ICLR 2023_

### Official Review · Reviewer_nhwb · 2022-10-25

**Confidence:** 4
**Correctness:** 4
**Technical Novelty And Significance:** 3
**Empirical Novelty And Significance:** 3
**Recommendation:** 6

**Clarity, Quality, Novelty And Reproducibility:**

The proposed method is novel. Some aspects of the method like selective attention maps and conditional adapters can be explained with more clarity. An average deep learning researcher who is not super familiar with the SSL field would find it different to understand. In terms of reproducibility, I didn't find the code that enables the reproducibility of the results. Authors are encouraged to share the code in the supplementary material. In terms of quality, the experiments and ablation studies are exhaustive and well-performed.

**Strength And Weaknesses:**

1. The proposed method to distill knowledge from an existing SSL model to build a stronger new model is novel and addresses an important issue of training/improving SSL models in resource-constraint situations.
2. Exhaustive experimentation shows that their approach outperforms existing approaches on varied vision tasks and at a fraction of computation time.
3. The contribution of different parts of the model like patch-norm features, selective attention maps and others are well demonstrated by the ablation experiments performed by the authors.

Weakness/Suggestions:
1. It would be nice to see the computation comparisons in terms of FLOPs.
2. I would recommend modifying the schematic diagram, especially the Attention Selection and Conditional Adapter part so that it is more intuitive from the schematic diagram itself. Currently, just from the schematic diagram, it is hard to understand these modules of the method.
3. Maybe in the appendix, it would be nice to visualize some more reconstruction targets (i.e. the patch-dim normalized features and selected attention maps) so that it helps to develop an intuition for the readers for what knowledge the new model is trying to distill.

**Summary Of The Paper:**

The paper introduces a novel method to train an SSL model by distilling knowledge from an existing SSL model (base model). By leveraging the base model, the method can help to learn a stronger SSL model in a cost-friendly way. Specifically, they propose two complementary reconstruction targets that enable knowledge distillation, 1)the patch-dim normalized features and 2) patch attention maps with rich semantics. They also have a conditional adapter module that adapts the new model's prediction to that of the base model. Exhaustive experimentation shows that their approach outperforms existing approaches on varied vision tasks and at a fraction of computation time.

**Summary Of The Review:**

Overall, based on the strengths, weaknesses, and other factors like clarity, novelty, and reproducibility, I place this submission marginally above the acceptance threshold (6).

---

> ### Author Response · Authors · 2022-11-16
> **Response to Reviewer nhwb**
>
> We highly appreciate your careful reading and comments, based on which we give point-to-point reply to address your concerns and also revise the manuscript accordingly.
>
> **Training cost comparison. (Q1)** We add the FLOPs, training time, and parameters comparison as shown in the table below.
>
> TEC requires shorter training time to achieve better performance than the base models.
> For example, TEC outperforms iBOT/MAE by 0.7%/1.1% Top 1 accuracy with only 7%/20% training time.
> As only a part of the model requires gradients in some SSL methods, e.g. the base model in TEC and online model in iBOT, which requires no backward cost to compute gradients, we compare the FLOPs for network parts with/without gradients.
> Benefiting from only processing the unmasked patches in the encoder and the shallow two-layer decoder,
> TEC requires smaller training FLOPs with gradients than iBOT and MAE.
> The extra FLOPs (FLOPs without gradients) of the base model in TEC are smaller than the online network in iBOT
> because no extra head is needed for the base model in TEC.
> Compared to MAE, the extra FLOPs of the base model can be partly balanced by the smaller FLOPs with gradients in TEC.
> Therefore, TEC has a similar training time with MAE for each training iteration.
>
>
> | Method   | Epoch | Time (8xA100) | FLOPs (with grad) | FLOPs (no grad) | Parameters | Top 1 acc. |
> | -------- | ----- | ------------- | ----------------- | --------------- | ---------- | ---------- |
> | VIT-B    | -     | -             | 17.6G             | -               | 86.6M      | -          |
> | iBOT     | 1600  | 361h          | 19.2G             | 19.2G           | 96.3M      | 84.1       |
> | TEC_iBOT | 300   | 25h           | 8.3G              | 17.6G           | 118.6M     | 84.8       |
> | MAE      | 1600  | 125h          | 9.8G              | 0G              | 111.9M     | 83.6       |
> | TEC_MAE  | 300   | 25h           | 8.3G              | 17.6G           | 118.6M     | 84.7       |
>
>
>
> **Schematic diagrams with more details. (Q2)** Due to limited space in the manuscript,
> we draw more detailed schematic diagrams of attention selection and conditional adapters in the appendix to facilitate understanding of these modules.
>
> **Visualization of reconstruction targets. (Q3)** We add the visualization of patch-dim normalized features and selected attention maps
> in the appendix.
> We show the visualization of patch-dim normalized feature-level targets of iBOT base
> model in Fig. 9 of the manuscript.
> Patch-dim normalized features are more distinguishable compared to the original
> and channel-dim normalized features.
> The spatial relation among feature patches is more clearly shown by the patch-dim normalization.
> We visualize the selected semantic attention-level targets from the iBOT base model in Fig.8 of the manuscript.
> The averaged attention maps of the class token mostly focus on the high-semantic objects, thus making the selected patches belong to the high-semantic objects.
> The attention maps of selected patches contain the semantic relation between high-semantic objects and other regions.
> Different patches have some unique attention parts that differ from other patches.
> The attention maps of these selected patches focus on similar semantic objects but
> are complementary in some parts,
> which explains why using attention maps of selected patches is better than only using class token attention maps in Table 6(b) of the manuscript.
>
>
> **Reproducibility.**
> We have submitted the supplementary which includes the source code to ensure reproducibility.
> We certainly will release our source code and pretrained models, since we expect future works to sustainably surpass ours with the help of our pretrained SSL model.

---

### Official Review · Reviewer_cZ41 · 2022-10-26

**Confidence:** 5
**Correctness:** 2
**Technical Novelty And Significance:** 2
**Empirical Novelty And Significance:** 2
**Recommendation:** 6

**Clarity, Quality, Novelty And Reproducibility:**

The presentation of the paper is good and well structured. However, the technical novelty of the paper is very limited. It simply combines different existing well known approaches such as adapters at input level, encoder level, distillation at feature level and attention level etc. The only contribution I see in the paper is the combination of these techniques for developing a technique for reusing pretrained SSL model.

**Strength And Weaknesses:**

**Strengths**:

- The paper is easy to read, and very well written with clear presentation of the figures/tables.
- The overall problem of reusing pretrained SSL models is interesting and the solution proposed to solve the problem has clearly demonstrated advantage on ImageNet dataset.

**Weaknesses**:

- The main technical contribution of the paper is actually on distilling knowledge from an existing pretrained model to another new model. This is in fact very much similar to reversed KD presented in "Revisiting knowledge distillation via label smoothing regularization" (transferring knowledge from a small model to a large model) but there is no discussion about this in the paper. I don't fully agree with the authors that knowledge distillation only focuses on transferring knowledge from strong and large models to compact new models. Authors should clearly discuss about the similarity between the proposed framework and (reversed) KD.

- Is there any restriction on the model size across base and new model? Since KD is agnostic to the model size, authors should perform experiments by transferring knowledge from a smaller SSL model for training a larger model, e.g., use of ViT-Small for training ViT-Base with MAE.

- In many practical scenarios, multiple pretrained models are usually available that can be leveraged for training a new model? How this framework can be used to exploit multiple models with same and/or different architectures? Authors should perform experiments to verify this using multiple base models.

- Can the proposed framework exploit SSL models trained using instance discrimination, e.g., SimCLR for training a new ViT MAE model? What about the performance in these scenarios?

- Sustainability usually refers to savings in terms of training cost which is measured by FLOPs or GPU wall time. While the comparison includes number of epochs, it does not reflect real savings. I would encourage authors to include Accuracy vs FLOPs and Accuracy vs Wall time to demonstrate how use of pretrained SSL model can help in training efficiency.  See this paper: Knowledge Inheritance for Pre-trained Language Models for more details which uses a smaller pretrained model to accelerate training of a larger language model or bert2BERT: Towards Reusable Pretrained Language Models which solves a similar problem in NLP.

- The proposed method uses a bunch tricks on top of the vanilla knowledge transfer strategy without analyzing other alternatives. What is the use of adapters at the input level and encoder level is not clear? Can authors qualitatively demonstrate how this helps in learning better features in the new model?

- What about the number of trainable parameters? What about the increase in training time due to the increase in number of parameters? A more thorough analysis on this should be included in the paper.

-  Besides empirical performance, how does semantic attention help learning diverse complementary features? Can authors perform visualization experiments to verify this?

- What about the additional computation incurred by passing all the images through the base model? An analysis on how much data we need to pass through the base model for an effective knowledge transfer should also be included in the paper.

- Did authors try considering this knowledge transfer as an additional regularizer on top of the standard MAE loss for the new model? What happens if we train the new model using a combination of the standard MAE loss that does depend on the base model and another loss that actually matches the prediction with the base model using the proposed framework?

**Summary Of The Paper:**

This paper presents a new reusable self-supervised learning framework by distilling knowledge from existing pretrained SSL models. Specifically, authors first introduce patch-relation enhanced targets to encourage the new model to learn semantic-relation knowledge and then introduce a conditional adapter that adaptively adjusts new model prediction to align with the target of each base model. Experiments using MAE and iBOT show the effectiveness of the proposed framework on ImageNet dataset.

**Summary Of The Review:**

The experiments are limited and not convincing in the current version of the paper. Due to the limited novelty and lack of convincing experiments, my initial recommendation is to reject the paper (below the acceptance threshold).

---

> ### Author Response · Authors · 2022-11-16
> **Response to Reviewer cZ41 (PART 3)**
>
> **TEC as an additional regularizer for MAE pretraining. (Q10)**
> We use the MAE and TEC decoders to predict RGB images and base model targets
> respectively.
> As shown in the table below, TEC can help the pretraining of MAE.
> But only using TEC for pretraining is much better than combining it with MAE.
> We assume that the prediction target of MAE enables the model to learn semantic-unrelated knowledge, thus slowing down the model convergence.
>
> | Method      | Epoch | Top 1 acc. |
> | ----------- | ----- | ---------- |
> | MAE         | 300   | 82.9       |
> | MAE         | 1600  | 83.6       |
> | TEC_MAE     | 300   | 84.7       |
> | MAE+TEC_MAE | 300   | 84.0       |
>
> **On technical novelty.**
> The proposed TEC is to learn and surpass pretrained SSL base models with various properties.
> However, directly distilling the knowledge from the base model to the new SSL model suffers from two issues that inhibit the above target.
> In this work, we propose novel solutions to the two issues in SSL setting, which are not considered in previous works. These two solutions actually differ our work from others, since our work is not a simple combination of previous techniques but designs some unique and elaborate solutions for SSL setting.
>
> Specifically, the technical novelty of TEC lies in two aspects, i.e. conditional pretraining and target enhancement.
> 1) We propose the target enhancement to stress the semantic relations of patch-level features from base models, benefiting the mask image modeling (MIM) learning from base models.
> The patch-dim normalized features and selected semantic attention maps
> are two simple implementations to achieve the goal of target enhancement.
> 2) We propose conditional pretraining to enable the adaptation of the new model
> to various types of base models.
> The input and encoder adapters are two simple yet effective modules for achieving conditional pretraining.
>
>
> [1] Revisiting knowledge distillation via label smoothing regularization
>
> [2] Knowledge Inheritance for Pre-trained Language Models
>
> [3] bert2BERT: Towards Reusable Pretrained Language Models

---

> ### Author Response · Authors · 2022-11-16
> **Response to Reviewer cZ41 (PART 2)**
>
> **Training cost comparison. (Q5, Q7, Q9)**
> Thanks for your suggestion.
> We have included these two related works [2,3] in NLP into the manuscript and followed them to add the FLOPs, training time, and parameters comparison as shown in the table below.
>
> 1. Training time and performance:
> TEC requires shorter training time to achieve better performance than the base models.
> For example, TEC outperforms iBOT/MAE by 0.7%/1.1% Top 1 accuracy with only 7%/20% training time.
>
> 2. Trainable parameters:
> TEC has a similar number of parameters with MAE, because the shallow decoder
> saves parameters while adapters increase parameters.
> Both TEC and MAE have a larger number of parameters compared to iBOT due to the extra decoder.
> But benefiting from the decoder,
> they only process the visible patches in the encoder, thus requiring a smaller training cost than iBOT.
>
> 3. Training FLOPs:
> As only a part of the model requires gradients in some SSL methods, e.g. the base model in TEC and online model in iBOT, which requires no backward cost to compute gradients, we compare the FLOPs for network parts with/without gradients.
> Benefiting from only processing the unmasked patches in the encoder and the shallow two-layer decoder,
> TEC requires smaller training FLOPs with gradients than iBOT and MAE.
> The extra FLOPs (FLOPs without gradients) of the base model in TEC are smaller than the online network in iBOT
> because no extra head is needed for the base model in TEC.
> Compared to MAE, the extra FLOPs of the base model can be partly balanced by the smaller FLOPs with gradients in TEC.
> Therefore, TEC has a similar training time with MAE for each training iteration.
>
> | Method   | Epoch | Time (8xA100) | FLOPs (with grad) | FLOPs (no grad) | Parameters | Top 1 acc. |
> | -------- | ----- | ------------- | ----------------- | --------------- | ---------- | ---------- |
> | VIT-B    | -     | -             | 17.6G             | -               | 86.6M      | -          |
> | iBOT     | 1600  | 361h          | 19.2G             | 19.2G           | 96.3M      | 84.1       |
> | TEC_iBOT | 300   | 25h           | 8.3G              | 17.6G           | 118.6M     | 84.8       |
> | MAE      | 1600  | 125h          | 9.8G              | 0G              | 111.9M     | 83.6       |
> | TEC_MAE  | 300   | 25h           | 8.3G              | 17.6G           | 118.6M     | 84.7       |
>
>
>
> **Effect of adapters. (Q6)** As introduced in Sec.2.2 of the manuscript,
> because SSL base models have various properties due to different training strategies, conditional pretraining using adapters is proposed to grant the new model with a stronger adaptation ability regarding a given SSL base model.
> We give the following evidence to demonstrate the effect of adapters:
> 1) Adapters have shown effectiveness by existing works in modulating mid-level features of the network for parameter-efficient fine-tuning on downstream tasks.
> Our work borrows the concept of adapters for parameter-efficient fine-tuning to make people better understand the goal of conditional pretraining.
>
> 2) The parameter-efficient finetuning results in Table 4 show that only finetuning adapters bring considerable performance gain, showing adapters do have the mid-level
> feature adaptation ability.
>
> 3) Figure 7 shows the encoder adapters gather features from different stages of the new model to be suitable for MAE/iBOT base model; i.e. iBOT base model requires adapters to provide more features from deeper layers, while MAE base model makes adapters focus more on shallow layers.
>
> 4) Ablation in Table 6(b) shows both input and encoder adapters help to improve the new model performance.
>
>
>
> **Visualization of semantic attention-level targets. (Q8)**
> We visualize the selected semantic attention-level targets from the iBOT base model in Fig.8 of the manuscript.
> The averaged attention maps of the class token mostly focus on the high-semantic objects, thus making the selected patches belong to the high-semantic objects.
> The attention maps of selected patches contain the semantic relation between high-semantic objects and other regions.
> Different patches have some unique attention parts that differ from other patches.
> The attention maps of these selected patches focus on similar semantic objects but
> are complementary in some parts,
> which explains why using attention maps of selected patches is better than only using class token attention maps in Table 6(b) of the manuscript.

---

> > ### Comment · Reviewer_cZ41 · 2022-12-05
> > **Final Response**
> >
> > I thank the authors for their response and effort on the new experiments. After carefully reading the authors response and other reviewers concerns, I am increasing my rating to 6 as the rebuttal clarified some of my concerns. However, the difference with (reverse) knowledge distillation is still not clear. Also, the additional computation incurred by passing all the images through the base model and an analysis on how much data we need to be passed through the base model for an effective knowledge transfer should also be included in the paper. Authors response to Training cost comparison. (Q5, Q7, Q9) is not clear. I would recommend authors to take only one base model and one target model to clearly analyze all the costs.

---

> > > ### Author Response · Authors · 2022-12-10
> > > **Follow up with Reviewer cZ41 (Part2)**
> > >
> > > **Data passed through the base model.** We mask different ratios of the input for the base model to reduce the data passed through the base model.
> > > The mask ratio for the new model is set to 75%.
> > > For easy implementation, only the patch-dim normalized features are used as the reconstruction target while the semantic attention maps are not used.
> > > As shown in the table below, the performance drops with the increase of mask ratio for base model input.
> > > Still, a large masking ratio of 75% for base model outperforms the MAE with a considerable gain.
> > > Therefore, masking input for base model can be used as a trade-off between training cost and performance in TEC.
> > >
> > >
> > > | Base      | Base Mask Ratio | Epoch | Top 1 acc. |
> > > |-----------|-----------------|-------|------------|
> > > | MAE-ViT-B |                 | 1600  | 83.6%      |
> > > | TEC*      | 0%              | 300   | 84.6%      |
> > > | TEC*      | 25%             | 300   | 84.5%      |
> > > | TEC*      | 50%             | 300   | 84.3%      |
> > > | TEC*      | 75%             | 300   | 84.2%      |
> > >
> > >
> > > **More clear explanation of training cost comparison.**
> > > We follow your suggestion to only compare one base model for clear understanding.
> > > We choose the iBOT base model, since it helps TEC achieve SOTA performance.
> > >
> > >
> > > 1. Training time and performance:
> > > On 8 A100 GPUs,
> > > iBOT requires 13.5 min for one training epoch, while TEC only required 5 min per training epoch.
> > > iBOT trained with 1600 epochs achieves 84.1% top1 accuracy, while TEC trained
> > > with 300 epochs reaches 84.8%.
> > > Therefore,
> > > TEC outperforms iBOT by 0.7% Top 1 accuracy with only 7% (25h vs. 361h) training time.
> > >
> > > 1. Training FLOPs:
> > > TEC and iBOT have the same inference cost, since only the ViT model is used for inference or downstream tasks.
> > > Since for pretraining, only a part of the model requires computing gradients in iBOT and TEC. For example, the base model in TEC and the online model in iBOT require no backward cost to compute gradients.
> > > To analyze the training cost, we divide the training cost into the network parts with/without gradients, since gradient computation is often much more computationally expensive.
> > > TEC requires much smaller training FLOPs with gradients than iBOT （8.3G vs. 19.2G), since TEC  only processes the unmasked patches in the encoder and uses a shallow decoder.
> > > The FLOPs without gradients of the base model in TEC are smaller than the online network in iBOT (17.6G vs. 19.2G), because no extra head is needed for the base model in TEC.
> > >
> > > 1. Trainable parameters:
> > > TEC and iBOT have the same inference parameters as only the ViT model is used for inference or downstream tasks.
> > > TEC has a larger number of training parameters compared to iBOT due to the extra shallow decoder and
> > > adapters (118.6M vs. 96.3M).
> > > But benefiting from the extra modules,
> > > TEC only processes the visible patches in the encoder, thus requiring a smaller training cost than iBOT. Moreover, for inference, TEC discards the decoder and thus enjoys the inference cost as iBOT because both use the same ViT encoder.
> > >
> > >
> > > | Method   | Epoch | Time (8xA100) | FLOPs (with grad) | FLOPs (no grad) | Trainable Parameters | Inference Parameters | Top 1 acc. |
> > > | -------- | ----- | ------------- | ----------------- | --------------- | ---------- | ---------- | ---------- |
> > > | iBOT     | 1600  | 361h          | 19.2G             | 19.2G           | 96.3M    | 86.6M | 84.1       |
> > > | TEC_iBOT | 300   | 25h           | 8.3G              | 17.6G           | 118.6M   | 86.6M  | 84.8       |

---

> > > ### Author Response · Authors · 2022-12-10
> > > **Follow up with Reviewer cZ41 (Part1)**
> > >
> > > Dear Reviewer cZ41,
> > >
> > > Thanks for your helpful suggestions. We have carefully prepared a response to address your concerns.
> > >
> > > **The difference with reverse knowledge distillation.**
> > > Sustainable SSL can be regarded as a special case of 'self-supervised distillation method' that the student model should outperform the teacher model in self-supervised setting.
> > > At the concept level, the major difference between sustainable SSL and reverse knowledge distillation is their different settings: the former focuses on self-supervised setting, while the latter targets at supervised learning setting. These different settings accordingly suffer from different distillation  problems and also require different solutions.
> > >
> > >  Specifically, reverse knowledge distillation is treated as 'a regularization term than similarity information of categories'[1]. So it highly relies on human annotation.
> > > In contrast, sustainable SSL has no available human annotation and the visualized results in Fig.2 of the manuscript show our TEC method makes the model capture all regions instead of
> > > a part of semantic regions. Moreover, TEC further needs to consider two problems in SSL distillation setting. The first one is how to handle different base models with various properties
> > > as self-supervised models are trained with various strategies. To solve it, we propose conditional pretraining which adaptively adjusts features of new model to be suitable for base model targets and thus helps relieve the issue. The second one is how
> > > to obtain targets with useful information from the base model
> > > as no classifier is available in self-supervised models. Thus we propose target enhancement that stresses the patch-level
> > > semantic relations to facilitate the mask image modeling learning scheme.
> > > Both conditional pretraining and target enhancement in TEC method significantly improves the performance. By comparison, reverse knowledge distillation considers the basic supervised settings of knowledge distillation and does not need to consider the above issues in SSL settings. So it only adds an extra loss term regarding teacher guidance to improve performance.
> > >
> > >
> > > [1] Revisiting knowledge distillation via label smoothing regularization

---

> ### Author Response · Authors · 2022-11-16
> **Response to Reviewer cZ41 (PART 1)**
>
> We highly appreciate your careful reading and comments, based on which we give point-to-point reply to address your concerns and also revise the manuscript accordingly.
>
> **Relation with distillation methods. (Q1)**
> Thanks for your suggestion. We discuss the relationship between sustainable SSL and distillation in more detail in the revision. Sustainable SSL can be regarded as a special case of 'self-supervised distillation method' that the student model should outperform the teacher model in self-supervised setting.
> Compared to Reversed KD [1] which conducts the distillation under a supervised setting, our work focus on a self-supervised setting.
> We have added the discussion regarding Reversed KD in the manuscript.
> We also compare our method with several recently proposed self-supervised distillation methods on the fully finetuning performance on ImageNet.
> The following table shows the remarkable improvement of TEC over other self-supervised distillation methods.
> For using MAE ViT-B as the base model, TEC outperforms FD by a noticeable gain of 0.9%.
> When comparing with MaskFeat which also applies the MIM scheme, TEC has a gain of 0.6% when using MoCov3 ViT-B as the base model.
>
>
> | Method          | Base      | Arch  | Epoch | Top 1 acc. |
> | --------------- | --------- | ----- | ----- | ---------- |
> | MAE             | -         | ViT-B | 1600  | 83.6%      |
> | FD_MAE          | MAE-ViT-B | ViT-B | 300   | 83.8%      |
> | TEC_MAE         | MAE-ViT-B | ViT-B | 300   | 84.7%      |
> | MoCov3          | -         | ViT-B | 300   | 83.2%      |
> | MaskFeat_MoCov3 | MoCov3-ViT-B | ViT-B | 300   | 83.9%      |
> | TEC_MoCov3      | MoCov3-ViT-B | ViT-B | 300   | 84.5%      |
>
>
> **Vit-Small as the base model for training Vit-Base. (Q2)**
> There is no restriction on the model size across the base and new models.
> We follow your suggestion to train a ViT-Base new model for 300 epochs under the guidance of 300 epoch MAE pretrained Vit-Small,
> and our TEC achieves 83.8 Top-1 acc, which outperforms the base model with a considerable gain of 3%.
>
> | Type       | Method         | Arch  | Epoch | Top 1 acc. |
> | ---------- | -------------- | ----- | ----- | ---------- |
> | Base model | MAE            | ViT-S | 300   | 80.8       |
> | New model  | TEC_MAE(ViT-S) | ViT-B | 300   | **83.8**   |
>
>
> **Multiple based models. (Q3)** We use two base models (MAE and iBOT) to train a new model.
> Our initial trial shows that the performance is similar to that of the model trained
> with one base model and no extra gain is observed.
> We will further explore making use of multiple base models in our future work.
>
>
> **Instance discrimination SSL model as the base model. (Q4)**
> We train the TEC model using the instance discrimination MoCov3 pretrained ViT-B as
> the base model, since our work currently focuses on ViT models.
> TEC outperforms the MoCov3 by 1.3% on Top-1 acc,
> showing that TEC can also learn and surpass instance discrimination based SSL models.
>
> | Type       | Method     | Epoch | Top 1 acc. |
> | ---------- | ---------- | ----- | ---------- |
> | Base model | MoCov3     | 300   | 83.2       |
> | New model  | TEC_MoCov3 | 300   | **84.5**   |

---

### Official Review · Reviewer_2pYM · 2022-10-29

**Confidence:** 3
**Correctness:** 3
**Technical Novelty And Significance:** 2
**Empirical Novelty And Significance:** 3
**Recommendation:** 5

**Clarity, Quality, Novelty And Reproducibility:**

In general, this paper is well-written, and the logistics are clear. The proposed new problem setting seems promising, but it needs more experiments and clear motivation. The proposed method is incremental, as it only involves designing new pretext tasks.

**Strength And Weaknesses:**

Strength:

1. The problem setting is new or interesting. It should find applications in different areas, such as medical images.

2. The proposed solution is reasonable and achieves good performance.

Weaknesses:

1. It seems that the authors should further clarify/motivate the proposed setting. Why do we need "sustainable" SSL. In the current introduction, the authors mention that researchers may have limited computational budgets to XXXXX, while it seems that the proposed solution cannot reduce the computation budget if I want to adopt a pretrained SSL model to downstream tasks. For my understanding, we still need to use a large ViT model and need to train many epochs. In other aspects, the previous SSL knowledge distillation can reduce the computational budget by re-training a small model.

2. For the imagenet fine-tuning experiments, it is hard for me to understand the motivation of this new problem setting. This experiment looks like we want to continue "pre-trained" an SSL model with large-scale datasets. In this case, why do we not adopt a "well-trained" SSL model (MAE trained with 1600 epochs)? As it is only pretrained once, it can be done by a giant company and release to others for downstream tasks. In a word, the authors should reconsider the evaluation part to support the argument for adopting this new problem setting.

3. It seems that the authors should compare with SSL knowledge distillation works. I understand the problem settings are different, while the method design should be similar. In some sense, the proposed method is also an advanced knowledge distillation method.

4. For Table 1, it seems that the authors should clearly mention how the previous results () are obtained. For example, from released public models or re-implemented models? And how the base SSL model is acquired for the proposed framework. Did the authors re-train the base MAE model, or did the authors adopt the publicly released MAE model?

5. The technical novelty may be incremental, as the key technical components are focused on designing new pretested tasks.

**Summary Of The Paper:**

This work studies a relatively new problem in self-supervised learning, which is called sustainable self-supervised learning. Based on a pretrained SSL model (or partially trained with certain epochs), the authors want to further improve it with the proposed target-enhanced conditional mask-reconstruction. The authors conduct experiments on benchmark datasets and shows the effectiveness of the proposed "fine-tuning" strategy.

**Summary Of The Review:**

In summary, this paper studies an interesting paper and should have an impact on the computer vision field, while more clarification of motivation and experimental evaluation is needed.

---

> ### Author Response · Authors · 2022-11-16
> **Response to Reviewer 2pYM (PART 2)**
>
> **Source of previous results. (Q4)** The previous results are quoted from the official reported results. The MAE/iBoT models we used are from their publicly released versions. We have added an explanation of the model source in the manuscript.
>
> **Technical novelty. (Q5)**
> The SSL pretraining is about designing new pretext tasks to conduct self-supervised learning. Indeed, related works (e.g. MAE, iBoT, Beit, and DINO) all focus on designing new pretext tasks.
>
> The proposed TEC is to learn and surpass pretrained SSL base models
> with various properties.
> However, directly distilling the knowledge from the base model to the new SSL model suffers from two issues that inhibit the above target. In this work, we propose novel solutions to the two issues in SSL setting, which are not considered in previous works. These two solutions actually differ our work from others, since our work is not a simple combination of previous techniques but designs some unique and elaborate solutions for SSL setting.
>
> Specifically, the technical novelty of TEC lies in two aspects, i.e. conditional pretraining and target enhancement.
> 1) We propose the target enhancement to stress the semantic relations of patch-level features from base models, benefiting the mask image modeling (MIM) learning from base models.
> The patch-dim normalized features and selected semantic attention maps
> are two simple implementations to achieve the goal of target enhancement.
> 2) We propose conditional pretraining to enable the adaptation of the new model
> to various types of base models.
> The input and encoder adapters are two simple yet effective modules for achieving conditional pretraining.

---

> ### Author Response · Authors · 2022-11-16
> **Response to Reviewer 2pYM (PART 1)**
>
> We highly appreciate your careful reading and comments, based on which we give point-to-point reply to address your concerns and also revise the manuscript accordingly.
>
> **Motivation of sustainable SSL. (Q1 and Q2)**
> The goal of sustainable SSL is to pursue stronger SSL pretrained models in an efficient manner with the help of the existing SSL models.
> The value of SSL pretraining is well acknowledged, but the high training cost restricts its development.
> We hope to encourage more researchers who are outside the 'giant company' and have no large computational budgets to explore better SSL pretraining schemes in a cost-friendly manner.
> Therefore, we need to find a way to reduce the computational cost required by SSL pretraining.
> The sustainable SSL provides a more cost-friendly solution to obtaining stronger SSL pretrained models based on existing ones,
> which greatly lowers the barrier to study the SSL field.
> For example, as shown in the following table (Table 6(e) in revision), our TEC outperforms the 1,600 epoch pretrained MAE model with a large margin when pretrained with only 100/300 epochs based on a 300 epoch pretrained MAE model, which greatly saves pretraining cost while achieving superior performance.
> Without sustainable SSL, one may need to pretrain a new SSL model from scratch with much more epochs to surpass the MAE model pretrained for 1,600 epochs.
>
> Regarding downstream tasks, sustainable SSL saves the pretraining budgets rather than the transfer learning cost on downstream tasks. We also believe that by using certain ways, e.g. sustainable pruning, the sustainable SSL idea might also save transfer learning costs,  and we will explore it in the future.
>
> | Method        | Epoch | Top 1 acc. |
> | ------------- | ----- | ---------- |
> | MAE           | 300   | 82.9       |
> | MAE           | 1600  | 83.6       |
> | TEC_MAE300EP  | 100   | 83.9       |
> | TEC_MAE300EP  | 300   | 84.3       |
> | TEC_MAE1600EP | 300   | 84.7       |
>
> **Evaluation for SSL pretrained models. (Q2)**
> Because the goal of sustainable SSL is obtaining stronger SSL pretrained models with the help of pretrained SSL models, we follow the standard evaluation settings in SSL pretraining works (e.g. MAE, iBOT, Beit, and DINO) to compare the Imagenet fine-tuning and transfer learning performance.
>
>
>
> **Comparison with distillation methods. (Q3)**
> We agree that our proposed TEC can be regarded as "an advanced knowledge distillation method".
> We compare our method with several recently proposed self-supervised distillation methods on the fully finetuning performance on ImageNet.
> The following table shows the remarkable improvement of TEC over other self-supervised distillation methods.
> For using MAE ViT-B as the base model, TEC outperforms FD by a noticeable gain of 0.9%.
> When comparing with MaskFeat which also applies the MIM scheme, TEC has a gain of 0.6% when using MoCov3 ViT-B as the base model.
>
>
> | Method          | Base      | Arch  | Epoch | Top 1 acc. |
> | --------------- | --------- | ----- | ----- | ---------- |
> | MAE             | -         | ViT-B | 1600  | 83.6%      |
> | FD_MAE          | MAE-ViT-B | ViT-B | 300   | 83.8%      |
> | TEC_MAE         | MAE-ViT-B | ViT-B | 300   | 84.7%      |
> | MoCov3          | -         | ViT-B | 300   | 83.2%      |
> | MaskFeat_MoCov3 | MoCov3-ViT-B | ViT-B | 300   | 83.9%      |
> | TEC_MoCov3      | MoCov3-ViT-B | ViT-B | 300   | 84.5%      |

---

### Official Review · Reviewer_uYmT · 2022-11-03

**Confidence:** 4
**Correctness:** 2
**Technical Novelty And Significance:** 3
**Empirical Novelty And Significance:** 3
**Recommendation:** 5

**Clarity, Quality, Novelty And Reproducibility:**

**Clarity**: the paper and figures is difficult to read and the contributions are not well stated. Also, I find that presenting the method in the scope of sustainability is rather unconvincing. The paper would benefit from some re-writing to emphasize more the fact that it is a kind of “self-supervised distillation method” (falling under the same umbrella as methods in the last paragraph of the related work section).

**Novelty**: The different components of the method are not novel: reconstructing dropped patches based with feature representations from another model as target (BeiT, MaskFeat), leveraging the salient attention maps given by a ViT (LeoPart), distilling from a pre-trained SSL model (clusterFit, maskfeat).
However we can argue that the novelty lies into their combination.

**Reproducibility**: As a practitioner, I would argue that the paper does not give enough implementation details for allowing the reproduction of the reported results (for example there is no implementation details about data augmentation nor about the downstream tasks). It wouldn't be a problem if the paper mentions that code will be made publicly available but there is no such mention in the current version.


**Strength And Weaknesses:**

**Strengths**

- Extensive comparisons with sota and thorough presentation of the related works.

- The experimental results and absolute numbers reported in this paper are strong. This paper advances the state of the art in self-supervised representation learning.

- In my opinion, the strongest result is in Table 7d, where the paper shows that MAE-300 epochs + TEC-100 epochs (i.e. 400 epochs)is competitive with MAE-1600 epochs pre-training. This convinces me that the method can speed up SSL pre-training.

- Many ablations are presented.


**Weaknesses**

- Unconvincing “sustainability” ability of the paper.
This work starts from the observation that even though SSL is progressing in computer vision, models are not really used in practice for downstream tasks. The paper argues that this is because (i) pre-training SSL costs are too high and (ii) new and better SSL methods are appearing very frequently, preventing the field to stabilize on a framework.
However this argumentation is flawed for the following reasons.
(i) Models with extremely high pre-training budgets are being adopted in practice (GPTs, CLIP, models trained on JFT), meaning that expensive pretraining costs do no prevent the pretrained models to be used subsequently.
(ii) The fact that new SSL methods arrive might just be a signal that SSL in vision is not fully mature yet and that further progress needs to be made to clearly beat supervised learning (which is still the most used pre-training in vision).

- Table 6 is the weakest part of this paper. It shows that TEC does not achieve “sustainability” as advertised in Figure 1.
Unlike the sustainability property advertised by the paper in Figure 1, Table 6 shows that a TEC model cannot be a based model for a new TEC-training round (marginal improvement of +0.1%).

- Table 1 should include comparison with strong supervised baseline like DeiT-3. In addition, this is very misleading in Table 1 not to count the base model epochs into the total number of epochs. In other words, when training TEC-MAE for 300 epochs, the column should be 1900 epochs (1600 + 300) instead of 300.

- Linear evaluation is 10 points below sota (iBOT for example). I find it weird that authors don’t report sota numbers in this Table 4.

- The method is complex with many added components (input and encoder adapters, use of self-attention maps for feature selection). Complexity might not go towards sustainability.

- This is not really a new SSL method but rather a feature distillation or enhancement one. Should compare with this literature (in last paragraph of related work) more and start from the same backbone as theirs for fair comparison.


**Summary Of The Paper:**

This paper proposes an unsupervised method for improving already trained self-supervised models. The method, TEC, consists in reconstructing the masked patch representations of an image. The target, i.e. the feature representation of the dropped patches, are obtained from a pre-trained SSL model.

**Summary Of The Review:**

For the reasons stated above (weaknesses section + problem of reproducibility), I lean towards rejection of the paper.

That being said, I still think the paper is promising and some reported results are strong. I will be willing to upgrade my initial assessment based on the authors’ rebuttal.

In any case (acceptance or rejection), I definitely think the paper should be re-written to tone down the “sustainability” aspect and present the contribution through the scope of “distillation” or “feature enhancing”.

---

> ### Author Response · Authors · 2022-11-16
> **Response to Reviewer uYmT (PART 3)**
>
> **Comparison with related works you mentioned. (Q in Novelty.)**
> We compare TEC and related works, and add this discussion to the revision.
> Similar to MIM works (e.g. BeiT[1], MaskFeat[2]), our TEC is also based on the MIM scheme.
> But to make sustainable SSL possible, we make TEC learn to surpass various pretrained SSL base models.
> Moreover, we propose target enhancement and conditional pretraining that enhance the MIM scheme to achieve this goal.
>
> Compared with LeoPart [3] which uses the attention map of the class token as the loss weights,
> we select patches with high semantics using class token and utilize attention maps of the selected patches as targets.
> Our selected semantic attention maps filter out possible noise and establish the correlation between the whole image semantic and the patch semantic. We show in Table 6(f) that using the attention maps of selected patches
> is better than solely using class token attention maps or all attention maps.
>
> Compared with ClusterFit [4], our method adopts different objectives and methodologies.
> ClusterFit conducts pretraining on the clustered pseudo-labels to reduce the overfitting
> of pretrained models to the pretext tasks. In contrast, our TEC aims to  outperform various pretrained SSL models with a target-enhanced conditional MIM scheme.
>
>
>
> **Comparison and discussion with distillation methods. (Q6)** Per your suggestion, we discuss the relationship between sustainable SSL and distillation in more detail in the revision. Sustainable SSL can be regarded as a special case of the 'self-supervised distillation method' that the student model should outperform the teacher model in a self-supervised setting.
> We have revised the title of our manuscript to *One step Towards Sustainable Self-supervised Learning*, and also rewritten the related content in the manuscript accordingly.
>
> We also compare several recently proposed self-supervised distillation methods on the fully finetuning performance on ImageNet.
> The following table shows the remarkable improvement of TEC over other self-supervised distillation methods.
> For using MAE ViT-B as the base model, TEC outperforms FD by a noticeable gain of 0.9%.
> When comparing with MaskFeat which also applies the MIM scheme, TEC has a gain of 0.6% when using MoCov3 ViT-B as the base model.
>
>
> | Method          | Base      | Arch  | Epoch | Top 1 acc. |
> | --------------- | --------- | ----- | ----- | ---------- |
> | MAE             | -         | ViT-B | 1600  | 83.6%      |
> | FD_MAE          | MAE-ViT-B | ViT-B | 300   | 83.8%      |
> | TEC_MAE         | MAE-ViT-B | ViT-B | 300   | 84.7%      |
> | MoCov3          | -         | ViT-B | 300   | 83.2%      |
> | MaskFeat_MoCov3 | MoCov3-ViT-B | ViT-B | 300   | 83.9%      |
> | TEC_MoCov3      | MoCov3-ViT-B | ViT-B | 300   | 84.5%      |
>
>
> **On Clarity.** Thanks for your suggestions. To improve clarity, we have revised the manuscript in the following aspects: 1) We revise the title of our manuscript to *One step Towards Sustainable Self-supervised Learning*. 2) We rewrite the relation between sustainable SSL and self-supervised distillation methods, in Sec.1, Sec.4, and Sec.5. 3) We add figures with more details and visualizations in the appendix for a better understanding of our method.
>
> **On reproducibility.**
> We have submitted the supplementary which includes the source code and also gives more details about data augmentation and downstream tasks. We certainly will release our source code and pretrained models, since we expect future works to sustainably surpass ours with the help of our pretrained SSL model. For pretraining, we follow MAE to use the basic random resize crop augmentation. For downstream tasks, we keep the exact settings with our baselines and only replace
> the pretrained weights.
>
> [1]Beit: Bert pre-training of image transformers
>
> [2]Masked feature prediction for self-supervised visual pre-training
>
> [3]Self-Supervised Learning of Object Parts for Semantic Segmentation
>
> [4]Clusterfit: Improving generalization of visual representations

---

> ### Author Response · Authors · 2022-11-16
> **Response to Reviewer uYmT (PART 2)**
>
> **Linear probing comparison. (Q4)** The following table and Table 9 in the revision report the accuracy of 1) learning probing and  2) adapter finetuning which only finetunes the adapter and the linear classifier for parameter-efficient finetuning.
>
> One can observe that 1) the linear probing performance of TEC relies on the base model, and 2) adapter finetuning significantly improves the performance. Indeed, most MIM-based models, e.g. BEIT and MAE, have much lower linear probing performance, since they do not use the global semantic learning loss, e.g. clustering loss or InforNCE instance discriminative loss. This also explains the lower performance of TEC compared with the global semantic learning methods, e.g. iBOT. But by finetuning the adaptors and also linear classifier, TEC improves iBOT with a remarkable margin of 3.9%. This is because as shown in Fig. 2 of the manuscript, iBOT focuses more on distinguishing the patches related to global semantics and ignores the semantics of other patches, while TEC can group the patches into several semantic groups and further identify the semantics of each group. In this way, finetuning adapters help activate the semantic groups that are related to global semantics required by the downstream tasks, thus improving the model's discriminability on global semantics and showing good parameter-efficient finetuning performance.
>
> | Method   | Epoch | Settings       | Top 1 acc. | Fully FT Top 1 acc. |
> | -------- | ----- | -------------- | ---------- | ------------------- |
> | BEiT     | 800   | Linear probing | 56.7       | 83.2                |
> | SimMIM   | 800   | Linear probing | 56.7       | 83.8                |
> | BootMAE  | 800   | Linear probing | 66.1       | 84.2                |
> | CAE      | 800   | Linear probing | 68.6       | 83.8                |
> | SemMAE   | 800   | Linear probing | 68.7       | 84.5                |
> | CMAE     | 800   | Linear probing | 73.9       | 84.7                |
> | Ge2-AE   | 800   | Linear probing | 75.3       | 84.8                |
> | MAE      | 1600  | Linear probing | 68.0       | 83.6                |
> | TEC_MAE  | 800   | Linear probing | 69.8       | 84.7                |
> | TEC_MAE  | 800   | Adapter FT     | 79.9       | 84.7                |
> | iBOT     | 1600  | Linear probing | 79.8       | 84.1                |
> | TEC_iBOT | 800   | Linear probing | 78.0       | 84.8                |
> | TEC_iBOT | 800   | Adapter FT     | **81.9**   | **85.1**            |
>
>
> **Technical novelty and simplicity of the proposed TEC. (Q5)**
> The proposed TEC is to learn and surpass pretrained SSL base models
> with various properties.
> However, directly distilling the knowledge from the base model to the new SSL model suffers from two issues that inhibit the above target. In this work, we propose novel solutions to the two issues in SSL setting, which are not considered in previous works. These two solutions actually differ our work from others, since our work is not a simple combination of previous techniques but designs some unique and elaborate solutions for SSL setting.
>
> Specifically, the technical novelty of TEC lies in two aspects, i.e. conditional pretraining and target enhancement.
> 1) We propose the target enhancement to stress the semantic relations of patch-level features from base models, benefiting the mask image modeling (MIM) learning from base models.
> The patch-dim normalized features and selected semantic attention maps
> are two simple implementations to achieve the goal of target enhancement.
> 2) We propose conditional pretraining to enable the adaptation of the new model
> to various types of base models.
> The input and encoder adapters are two simple yet effective modules for achieving conditional pretraining.

---

> ### Author Response · Authors · 2022-11-16
> **Response to Reviewer uYmT (PART 1)**
>
> We highly appreciate your careful reading and comments, based on which we give point-to-point reply to address your concerns and also revise the manuscript accordingly.
>
> **On the motivation of sustainable SSL. (Q1)** We strongly agree with your two points about SSL models, which are the exact reasons why we want to explore sustainable SSL.
>
> For (ii) that 'SSL in vision is not fully mature', instead of 'stabilizing on a framework', we hope to enable researchers to develop new and powerful SSL methods even if they do not have sufficient GPU resources and large-scale private data.
> By proposing sustainable SSL we aim to build a new stronger SSL model by using the existing pretrained SSL model instead of totally training a new one from scratch, and thus greatly relieves the computation cost and data resource requirement.
> For example, on ViT-B model, TEC trained for 800 epochs on ImageNet-1k (1.2M) achieves 85.1% accuracy on ImageNet-1k, i.e. a 2.2% improvement over CLIP pretrained with 32 epochs on a private 400 million text-image dataset and then fully finetuned on ImageNet-1k. This sustainable SSL greatly saves both computation and data cost. Indeed, without sustainable SSL, it is very hard to achieve such performance because most researchers do not have extremely high training budgets and large-scale private data required for CLIP model.
>
> For (i) that 'Models with extremely high pre-training budgets and large-scale private data (GPTs, CLIP, models trained on JFT) are being adopted in practice',
> this is true but due to limited computation and data resources, most researchers cannot easily improve these models even though they have better SSL algorithms at hand. Sustainable SSL makes it possible to develop stronger SSL models based on these expensive models with computational resources affordable to most researchers.
>
>
> **On results in Table 6.(Q2)** As we mentioned in the abstract and introduction of the manuscript,
> this work is an initial exploration of sustainable SSL to accelerate the learning speed and also improve existing pretrained SSL models.
> Our proposed TEC model at least demonstrates that a sustainable SSL strategy is possible and can surpass SOTA base models in a cost-friendly manner.
> As you observed, Table 6 shows that multi-round sustainable SSL is really hard. This is because in this work, for each round, multi-round sustainable SSL always uses the same learning strategy and thus is restricted to learning similar useful information, which limits the model performance.
> So in the future, we will focus on 1) how to evaluate the important missing properties of the current new SSL model, 2) how to find an SSL base model that has the important properties missed in the new SSL model, and 3) how to design a learning scheme for new models to learn these missing properties.
>
> To be more precise, we have revised the title of our manuscript to *One step Towards Sustainable Self-supervised Learning*, and also rewritten the related content in the manuscript accordingly.
>
> **Comparison with the strong supervised method Deit-3. (Q3)**  The following table (also Table 1 in the revision) compares our TEC with Deit-3, and shows the remarkable improvement of TEC models over
> Deit-3. Specifically, TEC_iBOT pretrained 800 epochs improves Deit-3 pretrained 800 epochs by 1.3%. When taking 300 epoch pretrained MAE as the base model, TEC with 100 pretraining epochs also clearly outperforms Deit-3 trained with 800 epochs, i.e. 400 training epochs in total to beat Deit-3 trained with 800 epochs, saving much computational cost.
>
> | Method         | Epoch | Top 1 acc. |
> | -------------- | ----- | ---------- |
> | Deit-3         | 800   | 83.2       |
> | TEC_MAE300ep   | 100   | 83.9       |
> | TEC_iBOT1600ep | 800   | **85.1**   |
>
> **Clarify pretraining epoch numbers. (Q3)** For fairness, we follow the previous works with extra guidance (e.g. SemMAE, FD-CLIP and MVP) to count the pretraining epochs  used by models for pretraining from randomly initialized weights. To avoid misunderstanding, we add the explanation in the caption of Table 1: *The pretraining epoch number of TEC denotes the one from randomly initialized weights under the guidance of base models, and does not include that of the base model.*

---

### Decision · Program_Chairs · 2023-01-20

**Decision:**

Reject

**Justification For Why Not Higher Score:**

The positioning of the paper should be improved. The proposed method allows to improve a pre-trained SSL backbone by exploiting it as a target predictor, and hence should be compared with distillation approaches. Some of the results are interesting, but they are not in axis with the claims about sustainability.

**Justification For Why Not Lower Score:**

N/A

**Metareview: Summary, Strengths And Weaknesses:**

This paper proposes a recipe for obtaining a stronger self-supervised model by exploiting and building upon an off-the-shelf SSL backbone. Using a form of MIM, the masked patches are represented using the pre-trained model. This is in line with some of the recent work where multimodal features are used in a similar fashion. The paper presents relatively strong results, especially when showing that 300 epochs of MAE pretraining + 100 epochs of TEC surpasses the 1600 epochs of pure MAE.
However, the proposed positioning about sustainability and motivation could be improved. Moreover, Table 6 shows that several rounds of TEC training do not improve results further, making the point of the paper weaker. Also, the difference between the proposed method and knowledge distillation is unclear. The paper has received borderline ratings, with a lot of constructive and actionable feedback. I encourage the authors to take this feedback into account, modify the paper’s positioning and submitting to another venue. For the time being, I recommend rejection.